# The Variation in the Traits Ameliorated by Inhibitors of JAK1/2, TGF-β, P-Selectin, and CXCR1/CXCR2 in the *Gata1*^low^ Model Suggests That Myelofibrosis Should Be Treated by These Drugs in Combination

**DOI:** 10.3390/ijms25147703

**Published:** 2024-07-13

**Authors:** Francesca Gobbo, Fabrizio Martelli, Antonio Di Virgilio, Elena Demaria, Giuseppe Sarli, Anna Rita Migliaccio

**Affiliations:** 1Department of Veterinary Medical Sciences, Alma Mater Studiorum University, 40126 Bologna, Italy; francesca.gobbo3@unibo.it (F.G.); giuseppe.sarli@unibo.it (G.S.); 2National Center for Drug Research and Evaluation, Istituto Superiore di Sanità, 00161 Rome, Italy; fabrizio.martelli@iss.it (F.M.); antonio.divirgilio@iss.it (A.D.V.); 3Department of Medical and Surgical Sciences, Alma Mater Studiorum University, 40126 Bologna, Italy; elena.demaria3@unibo.it; 4Altius Institute for Biomedical Sciences, Seattle, WA 98121, USA; 5Institute of Nanotechnology, National Research Council (Cnr-NANOTEC), c/o Campus Ecotekne, 73100 Lecce, Italy

**Keywords:** myelofibrosis, megakaryocytes, Gata1, TGF-β, P-selectin, IL-8

## Abstract

Studies conducted on animal models have identified several therapeutic targets for myelofibrosis, the most severe of the myeloproliferative neoplasms. Unfortunately, many of the drugs which were effective in pre-clinical settings had modest efficacy when tested in the clinic. This discrepancy suggests that treatment for this disease requires combination therapies. To rationalize possible combinations, the efficacy in the *Gata1*^low^ model of drugs currently used for these patients (the JAK1/2 inhibitor Ruxolitinib) was compared with that of drugs targeting other abnormalities, such as p27kip1 (Aplidin), TGF-β (SB431542, inhibiting ALK5 downstream to transforming growth factor beta (TGF-β) signaling and TGF-β trap AVID200), P-selectin (RB40.34), and CXCL1 (Reparixin, inhibiting the CXCL1 receptors CXCR1/2). The comparison was carried out by expressing the endpoints, which had either already been published or had been retrospectively obtained for this study, as the fold change of the values in the corresponding vehicles. In this model, only Ruxolitinib was found to decrease spleen size, only Aplidin and SB431542/AVID200 increased platelet counts, and with the exception of AVID200, all the inhibitors reduced fibrosis and microvessel density. The greatest effects were exerted by Reparixin, which also reduced TGF-β content. None of the drugs reduced osteopetrosis. These results suggest that future therapies for myelofibrosis should consider combining JAK1/2 inhibitors with drugs targeting hematopoietic stem cells (p27Kip1) or the pro-inflammatory milieu (TGF-β or CXCL1).

## 1. Introduction

Myelofibrosis (MF) is the most severe of the myeloproliferative neoplasms which are negative for the Philadelphia chromosome, which generates the BCR-ABL1 oncogene [1]. The complex features of MF include dysfunctions of the bone marrow, spleen, and blood. Bone marrow abnormalities are represented by hematopoietic failure, increased pro-inflammatory milieu, excessive deposition of extracellular matrix (fibrosis), increased osteoblast proliferation and bone formation (osteosclerosis), and increased microvessel density (neo-angiogenesis) [2,3]. Abnormalities of the spleen include splenomegaly and ineffective extramedullary hematopoiesis, and those of the blood include anemia, with tear drop poikilocytes, and thrombocytopenia, with mega-thrombocytes. The discovery that the disease is driven by mutations in the thrombopoietin (TPO) axis, which constitutively activates JAK2 [2], raised a notable amount of hope that the disease could be treated by JAK inhibitors. However, clinical experience with Ruxolitinib, the JAK1/2 inhibitor with the longest clinical history in patients with MF, indicates that these drugs effectively ameliorate the symptoms of MF and specifically reduce splenomegaly but do not halt the progression of the disease to its eventual fatal outcome [4,5,6]. To address the unmet clinical needs of MF, numerous studies have scrutinized its pathogenesis with the aim of identifying additional therapeutical targets by using surrogate human and animal models [7]. These studies have reached the consensus that the pathogenesis of MF is triggered by the products of abnormal megakaryocytes (MKs) [8,9], which remain immature, produce low levels of platelets (plts), and are responsible for establishing a pro-inflammatory milieu [10] characterized by high levels of P-selectin, TGF-β1, and interleukin-8 (IL-8), which drive the fibrosis observed in the bone marrow. In addition, the driver mutations sustain myelo-proliferation by reducing the levels of p27kip1, an inhibitor of cell proliferation [11,12], in the malignant hematopoietic stem/progenitor cells (HSCs/HPCs) [13]. All these pathways are druggable, and over the years, we, as well as others, have conducted pre-clinical validations of the efficacy of various drugs, targeting them by using animal models [7]. For these studies, our laboratory has consistently used the *Gata1*^low^ mouse model, which carries the hypomorphic *Gata1*^low^ mutation, which deletes the first hypersensitive site (HS1) of *Gata1* necessary for its expression in MKs [14,15]. *Gata1*^low^ MKs express abnormalities similar to those identified in MF, including high levels of TGF-β1 [16,17], CXCL1, the murine equivalent of human IL-8 [10], and P-selectin [18]; with age, they develop an MF phenotype which closely resembles that of the patients [19]. Moreover, *Gata1*^low^ HSCs/HPCs express low levels of p27kip1 [20]. During the course of pre-clinical studies, we targeted these pathways by testing the efficacy of Aplidin (a compound isolated from an algae also known as Plitipepsin which increases p27Kip1) [20], two TGF-β inhibitors (SB431542, a commercially available compound which inhibits ALK5, the first element of the signal transduction of the TGF-β receptor [17]), the TGF-β trap AVID200 developed by Forbius [21], an antibody neutralizing murine P-selectin (RB40.34, the murine equivalent of Crizanlizumab [22]), the JAK1/2 inhibitor Ruxolitinib (alone or in combination) [23,24], and the CXCR1/2 inhibitor Reparixin [23]. Unfortunately, while targeting the majority of these pathways has been extremely effective in reducing MF in this mouse model, the results of the clinical trials using the activator of p27kip1 Aplidin [13], P-selectin (Crizanlizumab, NCT04097821), and TGF-β1 (the TGF-β1/3 trap AVID200, NCT03895112 [21]) carried out to date have been modest. Based on this experience, the consensus in the field is that treatment of MF may require combination therapies. Looking at the pre-clinical data published by us based on the *Gata1*^low^ model with a fresh eye, great qualitative and quantitative variation in the responses of the animals to the tested drug so far was identified. It was then hypothesized that a comparison of the effects exerted by the various drugs in animals may guide the identification of the most suited combination therapies for MF patients. With this aim, the efficacy exerted by Aplidin, Ruxolitinib, SB431542, AVID200, RB40.34, and Reparixin in ameliorating the various MF traits expressed by the present model was herein compared. Support for the rigor of this comparison was provided by the fact that (1) all the drugs were tested on the same animal model in comparable stages of disease progression (similar age, the same number of females and males, and limited statistical difference in the values expressed by the vehicle groups used across treatments), (2) the duration of the treatments was comparable across studies, (3) the endpoints were evaluated with the same standardized protocols by the same operators, and (4) none of the drugs investigated induced signs of toxicity. Data across the experiments were compared by expressing the values observed in the treated group as the fold change (Delta) with respect to that observed in the corresponding vehicle group, and the statistical differences among the fold changes were determined by multiparametric analyses. The endpoints analyzed had been either already published or retrospectively evaluated for this study by using specimens stored in the authors’ tissue bank (see Table 1 for detail).

## 2. Results

### 2.1. Effects on Cellularity, Fibrosis, and Neo-Angiogenesis in Bone Marrow

Due to the underlying fibrosis, the total number of cells present in the femur of *Gata1*^low^ mice was lower than normal [25]. The treatments which increased (positive Delta with respect to vehicle) the cellularity of the femur were Aplidin, the TGF-β inhibitors (both SB431542 and AVID200), and RB40.34 in combination with Ruxolitinib (54 days) (Figure 1). The increases induced by these five treatments were significantly different from those induced by the other treatments, which either had no effect or reduced (negative Delta) the number of cells present in the femur.

The severe fibrosis detected in the bone marrow of 10–12-month-old *Gata1*^low^ mice [25] was significantly reduced (negative Delta) by all the treatments (Figure 2). The greatest reductions, however, were sustained by treatment with Reparixin for 20 days, followed by treatment with SB431542 for 54 days. All the other treatments, including the TGF-β trap, AVID200, exerted significantly more modest effects on fibrosis. In particular, the effect exerted by Ruxolitinib was modest. The fact that AVID200 was less effective than SB431542 was puzzling, since both drugs act on the same pathway, TGF-β. The different efficacy of the two drugs could be related to the slightly shortest duration of the AVID200 treatment (42 vs. 54 days); however, an alternative hypothesis, such as more effective inhibition of the pathway by a small chemical compound which targets the intracellular signaling pathway vs. an antibody which depletes TGF-β in the microenvironment, cannot be excluded.

Neo-angiogenesis in the bone marrow is one of the traits of MF patients conserved in *Gata1*^low^ mice [25]. To compare the effects exerted by the different treatments on the level of vessel density in this model, results already published [17,21,26] and data obtained retrospectively from samples stored in the mouse tissue bank (Table 1 and Figure 3A,B) were used. With the exception of AVID200, all the treatments were equally potent in reducing vessel density in bone marrow from *Gata1*^low^ mice (Figure 3D).

### 2.2. Effects on Deposition and Maturation of Bone

Although osteopetrosis represents an important MF trait which is included in the morphological criteria used to assess disease progression [27,28], limited information is available regarding the pathobiological pathway leading to the osteopetrosis observed in this disease [29]. It has been previously described that the bone marrow from *Gata1*^low^ mice expresses pro-inflammatory cytokines (TGF-β1), growth factors (osteocalcin, platelet-derived growth factor [PDGF], and vascular endothelial growth factor [VEGF]), bone morphogenic proteins (BMP-2, -4, and -6 and their receptors BMPR-IA and -II), and bone-specific matrix proteins (osteonectin, bone sialoprotein, and osteopontin), some of which are produced by malignant MKs, known to induce osteosclerosis at levels greater than normal, already at 1 month of age [25,30,31]. These previous studies also demonstrated that the femoral bone of these mice remains immature with poor Ca^++^ deposition, a trait which is consistent with the osteopetrosis observed in MF patients [29,32]. To gain additional insights into the pathobiological pathway(s) leading to osteopetrosis in the present model, a computer-assisted method was devised to more accurately assess the osteosclerosis expressed by *Gata1*^low^ mice (Figure 4A). By using this method, it was first confirmed that the femurs from the mutant mice contained more bone than wild-type mice of comparable age and sex (Figure 4B). This method was then used to measure the area occupied by the bone in the femurs of all the treated mice by using photos stored in our slide bank. The results were then expressed as Delta changes between the treated group and the corresponding vehicle group (Figure 4C,D). Overall, none of the treatments effectively reduced the area of the femur occupied by bone in the present model. The modest reductions (negative Delta) observed with RB34.40 and Ruxolitinib in combination on day 54 and with AVID200 were not statistically significant.

Myelofibrosis is not only associated with increased bone deposition but also with reduced bone maturation [29]. In previous studies, we determined by means of Mallory trichrome staining that the maturation of the cortical and trabecular bones of the femur in *Gata1*^low^ mice is also reduced [26]. In fact, as expected, the cortical and trabecular bones of wild-type mice showed numerous areas stained in red, corresponding to mature lamellae rich in both collagen fibers and Ca^++^ (Figure 5A). By contrast, the corresponding areas from the *Gata1*^low^ femurs contain large fields stained in blue, which correspond to unmineralized osteoids, while fields stained in red are scanty. When the published effects of the treatments with RB34.40 and Ruxolitinib, alone or in combination, on the maturation of the bone were analyzed, the authors had already noticed that Ruxolitinib alone improved bone maturation [26]. The effects of SB431542, AVID200, and Reparixin on the ossification of bone from *Gata1*^low^ mice was specifically assessed for this study. The results indicate that AVID200 and Reparixin improved bone ossification since they restored the levels of mature lamellae (stained in red) present both in the cortical and trabecular areas of the *Gata1*^low^ femurs (Figure 5A,B). However, when the Delta values were compared, there were no significant differences in the improvement in bone maturation induced by the different treatments (Figure 5C,D).

### 2.3. Effects on Size and Fibrosis of Spleen

Extramedullary hematopoiesis and fibrosis in the spleen are serious complications of MF and lead to splenomegaly, a trait which greatly contributes to the malaise experienced by these patients [33]. *Gata1*^low^ mice also experience extramedullary hematopoiesis and fibrosis in the spleen, which profoundly alters the architecture of this organ [25,26]. The various treatments tested had various effects on the size and fibrosis of the spleen in this model. Spleen size was determined in terms of both weight and cellularity. Apart from Aplidin, AVID200 for 42 days, and Reparixin for 37 days, all the other treatments tested effectively reduced spleen weight. The greatest reductions were, however, exerted by SB431542 and Ruxolitinib, alone or in combination with RB40.34 (Figure 6). The reduction in spleen weight observed with Ruxolitinib in this mouse model was consistent with the important reduction in the splenomegaly induced by this drug in patients with MF [34].

By contrast, all the treatments tested had modest effects on the cellularity of the spleen compared with the corresponding vehicle (Figure 7). The authors believe that after treatment, the number of cells in this organ remained high despite the considerable reduction in weight, reflecting the fact that fibrosis, which is also manifested in the spleen [9], by reducing the space available for the cells, represents a confounding factor in determining spleen size by cell number.

Although all the treatments tested reduced (negative Delta relative to the vehicle) fibrosis in the spleen (Figure 8), the greatest reductions were observed in mice treated with Ruxolitinib alone or in combination with RB40.34, and they were significantly greater than those induced by all the other treatments.

### 2.4. Effects on TGF-β and CXCL1 Contents in Bone Marrow

In previous studies [10], it was shown that the plasma levels of pro-inflammatory cytokines, including those of TGF-β1 and CXCL1, did not correlate with their bioavailability in the microenvironment of bone marrow from *Gata1*^low^ mice and that in this model, it was microenvironmental bioavailability rather than systemic levels which is associated with fibrosis. More specifically, based on immunohistochemistry, the authors have previously reported that the bone marrow contents of TGF-β1 and of CXCL1 and its receptors (CXCR1 and CXCR2) were higher in *Gata1*^low^ mice compared with their wild-type littermates [10]. Moreover, these studies also demonstrated that the cells responsible for the accumulation of these pro-inflammatory proteins in the bone marrow microenvironment were the MKs [10].

To provide mechanistic insights into the efficacy of the various treatments tested on the myelofibrotic phenotype of *Gata1*^low^ mice, their effects on the TGF-β1 and CXCL1 contents in bone marrow were compared (Figure 9 and Figure 10).

The levels of TGF-β mRNA expressed in the bone marrow of the mice treated with SB431542 were previously determined by using real-time polymerase chain reaction (RT-PCR). These determinations indicated that SB431542 markedly reduced (by 4.2 times) the levels of mRNA expressed in this organ and, therefore, presumably also its protein content [17]. The effects of AVID200, RB40.34, Ruxolitinib, and Reparixin on the TGF-β content in bone marrow was instead previously assessed by using immunohistochemistry. These studies indicated that AVID200, Ruxolitinib (alone or in combination with RB40.34), and Reparixin all decreased TGF-β content in bone marrow [21,23,26] (Figure 9A–D). When these results were expressed as Delta of the corresponding vehicle, it was found that the greatest reductions were induced by AVID200, followed by Reparixin (day 37), while the effects exerted by RB40.34 and Ruxolitinib, alone and in combination, were modest (Figure 9F).

The CXCL1 content has previously been assessed only in research on RB40.34 and Ruxolitinib alone or in combination and Reparixin. These studies demonstrated a decrease in CXCL1 only in the bone marrow of the mice treated with RB40.34 in combination with Ruxolitinib [26]. For comparative purposes, a retrospective CXCL1 immunohistochemistry analysis of mice which had been treated with SB431542 and AVID200 was carried out, using samples stored in the tissue bank (Figure 10A–D). These analyses indicated that both these TGF-β inhibitors did not affect the content of CXCL1 in the bone marrow (Figure 10).

### 2.5. Effects on Platelet Counts

The blood of 10–12-month-old *Gata1*^low^ mice contains normal hematocrit (HCT) and white blood cell levels but has platelet levels lower than normal [25]. None of the treatments had any effect on the HCT and white blood cell counts of *Gata1*^low^ mice, which remained within normal ranges [17,20,21,23,24,26]. The majority of the treatments did not affect platelet counts (Figure 11), which, in general, remained low with the exception of the animals treated with Aplidin, SB431542, and more modestly, AVID200, in which the platelet counts were instead increased.

### 2.6. Effects on Megakaryocyte Maturation

The platelet counts in the blood of *Gata1*^low^ mice is low because the maturation of the MKs, the cells responsible for their production, is hampered by the reduced expression of GATA1, the transcription factor required for their maturation, sustained by the hypomorphic mutation [15,35]. To mechanistically understand why the platelet counts had been restored only by the TGF-β inhibitors, the GATA1 content in the MKs in bone marrow of the mice undergoing the various treatments was evaluated.

In the study published regarding SB431542 [17], the levels of GATA1 and the maturation state of the MKs in bone marrow was assessed by using quantitative RT-PCR and electron microscopy. The data showed that this TGF-β inhibitor restored the levels of GATA1 mRNA and normalized those for several maturation markers (reducing the expression of GATA2, Acetylcholinesterase, platelet factor 4, and P-selectin while increasing that of the von Willebrand factor). It also restored the ultrastructure features of the cells the cytoplasm of which contained well-developed platelet territories [17]. In the studies regarding RB40.34 and Ruxolitinib, alone or in combination, and Reparixin, the GATA1 content of the MKs in the bone marrow was determined by using confocal microscopy with antibodies against the lineage markers CD42b and GATA1 [23,26]. In order to compare the results of the treatments, a dedicated immunofluorescence analysis with the CD42b and GATA1 antibody of the bone marrow from mice treated with AVID200 and SB431542 was carried out (Figure 12A). The results across the treatments were then compared by expressing the frequency of GATA1-positive MKs as fold change with respect to the values observed in the vehicle group. None of the treatments affected the frequency of the MKs, which remained greater than normal (Figure 12B,C). The treatments which increased GATA1 content in the MKs the most were AVID200 (42 days) and Ruxolitinib (54 days), while more modest increases were observed with RB40.34 alone (54 days) and Reparixin (20 days) (Figure 12D–F). However, a closer look at the morphology of the GATA1-positive MKs indicated that in the case of the majority of treatments, the positive cells retained an immature morphology with monolobated nuclei and reduced cytoplasm (Figure 12G). By contrast, and as expected based on the electron microscopy observations published [17], the GATA1-positive MKs from SB431542-treated mice had a mature morphology with polylobated nuclei (Figure 12A,G). The increase in polylobated mature GATA1-positive MKs in mice treated with TGF-β inhibitors is consistent with the known inhibition exerted by this factor on MK maturation [36,37] and provides an explanation for the increased platelet counts observed in the blood from these mice.

## 3. Discussion

The drugs studied in this paper all had limited toxicity in the present model, as indicated by (1) the very limited number of deaths recorded for the entire duration of the treatments (one–two months), (2) the absence of changes in mobility or in the luster of the coat, and (3) the absence of anemia, one of the most common side effects of anti-cancer treatments. This is impressive since the mice analyzed in this study were old and, like MF patients, fragile. There were only two cases in which the procedure of drug administration had to be changed to account for the fragility of the model; in the first experiment with Aplidin^®^, the mice developed reduced mobility and drinking behavior, which were fully addressed in the following two experiments by keeping the mice hydrated by injecting 200 µL of sodium chloride (0.9% *w*/*v*) subcutaneously [20]. In the first experiment with AVID200, there was a 50% mortality rate in both the treated group and in the irrelevant IgG controls, which was addressed in the two additional experiments by reducing the frequency of the treatment from three to two times a week, which apparently allowed these mice to better cope with high levels of IgG in the blood. These considerations suggest that it is unlikely that the drugs used in these studies exerted significant off-target effects which may have confounded assessing the qualitative and quantitative differences in their efficacy on the MF traits of the present model. In addition, the fact that with very few exceptions, the values observed in the vehicle groups were very similar across the experiments ensured that the mice were always studied at similar stages of disease progression. The authors also believe that it is unlikely that differences in the vehicle (DMSO or saline) used to solubilize the drugs or in the way of administration (intraperitoneal vs. intravenous vs. gavage) may have represented confounding factors in evaluating the results.

In reading the pre-clinical data published regarding the *Gata1*^low^ model, a great qualitative and quantitative variation in the response of the animals to the various drugs tested was noted (Table 2). In order to quantify whether the similarities and the differences in the effects exerted by the various drugs on the MF traits displayed by the present model were statistically different, the phenotype of the treated mice was compared with that of the respective control group. The results are summarized in Table 3. How this variation could suggest drug combinations more suited to targeting the various facets of the disease will be discussed below.

It is reassuring that for those drugs which have already proceeded to clinical trials, the effects exerted on the model correlated well with those observed in the patients. In fact, it is impressive that in the present model, the drug most effective in reducing splenomegaly was Ruxolitinib, with splenomegaly being a feature which is consistently ameliorated by this drug also in MF patients [38,39]. The fact that the effects of all the other drugs on splenomegaly were inferior to those exerted by Ruxolitinib indicated that Ruxolitinib, or one of the new generation of JAK inhibitors under development, remains the first front-line treatment and should be included in all the combination therapies to be tested on MF patients in the future. Ruxolitinib was also one of the most effective drugs in reducing fibrosis in the spleen, while its effects on fibrosis in bone marrow were modest, suggesting that different cell populations are responsible for the fibrosis developed by the two organs. The hypothesis that different cell populations, activated by different mechanisms, are responsible for the fibrosis in the two organs was already suggested by a previous study which compared the bioavailability and signaling abnormalities of TGF-β in bone marrow and spleen from *Gata1*^low^ mice with respect to the controls [17]. Although both the bone marrow and spleen from the mutant mice expressed levels of TGF-β 1.2-fold greater than normal, the downstream TGF-β signature activated in the two organs and how they were affected by treatment with SB431542 were clearly different. The signature of the bone marrow included the activation of osteoblast differentiation, apoptosis, G1 arrest, and ubiquitin-mediated proteolysis; all these abnormalities were ameliorated by treatment with SB431542. By contrast, the signature of the spleen included only apoptosis and G1 arrest; this signature remained altered upon SB431542 treatment, which even induced additional abnormalities in the ubiquitin-mediated proteolysis pathway [17]. The cell populations responsible for the fibrosis in the bone marrow and spleen of the present model will be investigated as the subject of separate studies.

Mechanistically, the limited effects of Ruxolitinib on fibrosis in bone marrow could be related to its modest effects on the TGF-β and CXCL1 contents in this organ, which remained elevated in the treated group. Ruxolinitib, in combination with the antibody against the murine P-selectin RB40.34, was more effective than either of the two drugs alone in reducing CXCL1 content in bone marrow and had some effect on osteosclerosis. Based on these results, Novartis sponsored a clinical trial (NCT04097821) using the human P-selectin antibody Crizanlizumab, which was effective in reducing pain crises in patients with Sickle Cell Anemia [22] and MF. However, the company prematurely interrupted the trial when it lost interest in the clinical development of the drug. This lack of interest was based on studies which questioned the efficacy of Crizanlizumab on Sickle Cell Disease [40,41] and which led the European Medical Agency to revoke its authorization for the clinical use of this drug in Europe [42]. However, the clinical development of P-selectin inhibitors is still an active area of research, and new products could become available in the near future. The fact that Ruxolitinib in combination with the P-selectin antibody was more effective than either of the two drugs alone in the *Gata1*^low^ model suggests that future combination therapies should consider combining Ruxolitinib with drugs targeting inflammation. Interestingly, the same conclusion was reached by using the *JAK2*V617 mouse model, in which Ruxolitinib effectively reduced bone marrow fibrosis when used in combination with an epigenetic modifier targeting the *HMGA1* gene, which also reduced the pro-inflammatory milieu of the organ [43].

The TGF-β and CXCL1 inhibitors are two anti-inflammatory agents which target abnormalities with important pathobiological effects on the manifestation of MF and represent the most logical candidates to be used in combination with Ruxolitinib to treat MF. Although they were not tested in combination with Ruxolitinib in the present model, the careful comparison of their effects as monotherapy in the same pre-clinical model may not resolve the translatability issues concerning their therapeutic efficacy but may provide some rationale for prioritizing their investigation of combination therapies.

The inhibition of TGF-β signaling reduced bone marrow fibrosis and was the only treatment which increased platelet counts. Overall, the effects of AVID200 on the animal model were very similar to those observed in the subsequent clinical trial, just published (NCT03895112), which was conducted on a limited number of patients with advanced MF [44]. The results of the clinical trial indicate that although ineffective in preventing disease progression, AVID200 effectively increased platelet counts in the majority of the patients recruited. It is interesting that in the present model, the direct inhibition of TGF-β signaling with a small compound was more effective than reducing the TGF-β content with the AVID200 trap (Table 2). Unfortunately, although a clinical-grade compound, Galunisertib, which is structurally similar to the commercially available SB431542 used in the authors’ studies, has been developed by Lilly [45], the company has not so far been interested in studying this drug in MF. Therefore, to pursue this observation further, a collaboration was established with a medicinal chemist, Dr. Antonello Mai, who will use the innovative dual-target approach [46] to develop drugs which will affect JAK1/2 and AKL5 at the same time and which are eventually to be used instead of the combination of the two individual inhibitors to treat MF.

Reparixin was by far the drug which more effectively reduced fibrosis in bone marrow from *Gata1*^low^ mice (Table 2). This drug is currently under clinical evaluation in patients with advanced MF (MPN-RC 120, NCT05835466), and the trial has already recruited four patients. It is conceivable that if the results of this first trial are promising, there will be a second trial which will analyze the effects of Reparixin in combination with JAK inhibitors.

With the exception of AVID200, all the drugs were equally effective in reducing vessel density in bone marrow. This was not surprising since the increased vessel density observed in this disease could be mediated not only by high levels of VEGF, which the authors have previously shown to be normalized by inhibitors of TGF-β signaling [17], but also by CXCL1. The effects of CXCL1 may be either indirectly mediated, by reducing TGF-β content, or directly mediated, as is the case of epithelial malignancies in which high levels of CXCL8 enhance neo-angiogenesis and extracellular matrix remodeling to promote a microenvironment conducive for additional tumor growth and metastasis [47].

With the exception of RB40.34 in combination with Ruxolitinib, none of the drugs tested in this study reduced osteopetrosis. However, some improvement in bone maturation was observed with SB431542 and Reparixin; this effect was more pronounced on compact than on trabecular bone. This observation suggests that although these drugs do not activate the osteoclasts necessary to stimulate bone resorption, they do normalize osteoblast metabolism, preventing new-bone deposition. Further studies are necessary to clarify the role played by TGF-β and CXCL1 in the development of osteopetrosis in MF.

Osteoblasts and endothelial cells are elements of the endosteal and vascular niche which promote the functions of HSCs in bone marrow [48,49,50,51,52]. The endosteal niche is effective in juvenile mice, while the vascular niche provides support for HSCs in adult and senescent mice [53,54,55,56,57]. These niches affect HSC fate both directly, by secreting factors such as stem cell factor (SCF) and C-X-C motif chemokine ligand 12 (CXCL12), necessary for their survival, proliferation, and mobilization, and indirectly, by recruiting other cells, such as MKs, macrophages, and other stromal cells, which are responsible for secreting factors such as platelet factor 4 (PF-4; also known as CXCL4) and TGF-β, which force HSCs into quiescence, allowing them to retain stemness. The authors have previously shown that in old *Gata1*^low^ mice, HSCs are localized in the femur diaphysis in areas surrounded by microvessels, neo-bones, and MKs, while in their wild-type littermates, HSCs are localized in the epiphysis of the femur near trabecular bone [58]. The issue as to whether drugs affect microvessel density and/or bone deposition in mouse models has implications for their effects on the abnormalities of the “niches” which supposedly increase HSC trafficking, favoring extracellular hematopoiesis in MF. Although the authors’ studies did not characterize in detail the effects of the treatments on the hematopoietic niche, in the case of the P-selectin inhibitor and the TGF-β inhibitor SB431542, they have documented that these two drugs increase HSC content and restore hematopoiesis in bone marrow, suggesting that they normalize the functions of the HSC niche. By using another animal model, Yao et al. reached a similar conclusion on the effects exerted by TGF-β on the hematopoietic niche in bone marrow [59].

There is a discrepancy in the field regarding the criteria for assessing the effectiveness of a drug in reducing fibrosis in animal models and in patients. The criteria used more often in the animal models are based on computer-assisted quantification of reticulin content in bone marrow sections. By contrast, the criteria used to quantify fibrosis in patients are based on a semi-quantitative scale which includes both the complexity of the fibers and the presence of osteosclerosis [28]. The fact that in the present model, none of the drugs reduced bone deposition makes uncertain whether their efficacy in reducing fibrosis would remain valid if it were assessed by using the semiquantitative scale used for the patients. In this regard, the effects of Reparixin on bone marrow fibrosis were more limited when, in the Mathematical Programming Language (MPL) model, they were determined by using a semiquantitative scale [60]. A similar discrepancy between drug effectiveness in animal models and in patients has been observed in the field of idiopathic pulmonary fibrosis and was addressed by the American Thoracic Society by organizing a workshop dedicated to the formulation of consensus criteria to assess drug efficacy in animal models [61]. The time is right for scientists in the MF field to also organize a workshop dedicated to establishing consensus criteria to also assess drug efficacy in animal models for this disease.

One last point to discuss is that both Ruxolitinib and AVID200 had modest effects on the allele burden of the patients [4,21,38,39,62]. These results suggest that both drugs are ineffective in reducing malignant HSCs, which should be the ultimate goal for any cure of MF. Aplidin was the only drug included in the present panel which targeted abnormalities of malignant HSCs. In the *Gata1*^low^ model, Aplidin increased platelet counts and marrow cellularity and reduced microvessel density and the expression of TGF-β, vascular endothelial growth factor, and thrombopoietin [20], suggesting that the drug effectively normalized *Gata1*^low^ HSCs. Unfortunately, when the drug was tested on a limited number of patients with MF (11), although well tolerated with limited toxicity, it induced some response (improvement in anemia) in only one patient [13]. Aplidin activates p27Kip1, which is upstream and negatively regulates the activity of the cell cycle inhibitors RAC1/2 [63]; RAC1 inhibitors are emerging as an interesting bullet to treat various forms of cancer [64]. The present results suggest that the use RAC1/2 inhibitors, possibly in combination with Ruxolitinib, for the treatment of MF should be re-evaluated.

## 4. Materials and Methods

### 4.1. Mice

*Gata1*^low^ mice develop a myelofibrotic phenotype at 8–10 months of age which summarizes all the abnormal traits of the disease in humans. These mice represent a validated model for MF and are commercially available from the Jackson Laboratory (Cat. No. 004655; Bar Harbor, ME, USA). The animals used in these experiments were bred in the animal facility of the Higher Institute of Health, as previously described [25]. Littermates were genotyped at birth by using PCR, and those found not to carry the mutation were used as WT controls. The genotype was again confirmed before an animal was entered into an experimental group. The original experiments were carried out according to protocol D9997.121 approved by the Italian Ministry of Health on 2 September 2021 and according to European Directive 86/609/EEC. The current study was not subject to approval because it was conducted on published data and on samples stored in the tissue bank.

### 4.2. Mice Treatments

The treatments are described in detail in the original publication and are summarized in Figure 13 for clarity. All the experimental groups included equal numbers of male and female mice. No difference was observed in the outcomes between the two sexes; therefore, sex was not considered to be an independent variable.

Aplidin [20]: Aplidin^®^ was provided by Pharma Mar S.A., Colmenar Viejo, Madrid, Spain. The *Gata1*^low^ mice (10 months old) received either Aplidin^®^ (100 µg /kg/day) or the vehicle injected intraperitoneally (i.p.) for four cycles of 5 consecutive days 21 days apart and were then sacrificied for histopahological analyses. The mice were treated for a total of 20 days, and the duration of the treatment was 83 days.

SB431542 [17]: The *Gata1*^low^ mice (9 months old) were treated with SB431542 (Cat. No. S4317-5GM; Sigma-Aldrich, St. Louis, MO, USA), an inhibitor of the tyrosine kinase activity of TGF-β1 receptor type I. The mice were injected i.p. with SB431542 (60 mg/kg per day) or the vehicle (same volume) for 2 cycles of 5 consecutive days 2 day apart; they rested for 1 month and were then treated for 2 additional cycles (54 days in total). The mice were treated for a total of 20 days, and the duration of the treatment was 54 days.

AVID200 [21]: *Gata1*^low^ mice (10–12 months old) were randomly divided into 3 groups which were treated with either the vehicle (Irrelevant IgG) or AVID200 (5 mg/Kg twice a week i.p.) for 15 and 42 days with the TGF-β1/TGF-β3 protein trap. The mice were treated for a total of 13/21 days, and the duration of the treatment was 42/72 days total.

P-selectin inhibitor and Ruxolitinib, alone and in combination [24,26]: The *Gata1*^low^ mice (11 months old) were randomly divided into four groups which were treated as follows: Group 1: vehicle (2% *v*/*v* DMSO by gavage, negative control for Groups 3 and 4); Group 2: biotin-conjugated rat anti-mouse CD62P (RB40.34; Cat. No. 553743; BD Pharmigen, San Diego, CA, USA; 30 μg/mouse per day × three days per week by intravenous injection and then ip); Group 3: Ruxolitinib (Novartis Pharma AG, Basel, Switzerland; 45 mg/Kg twice a day 5 days a week by gavage); Group 4: biotin-labeled RB40.34 and Ruxolitinib in combination [26]. The experiment was conducted with two endpoints, on day 12 and day 54. The mice were treated for a total of 21/25 days, and the duration of the treatment was 54 days in total.

Reparixin [23]: The *Gata1*^low^ mice (8 months old) were implanted subcutaneously with ALZET^®^ Osmotic Pumps (model 2002) pre-filled with 200 μL of vehicle (sterile saline) or Reparixin (7.5 mg/h/Kg in sterile saline) as described by the manufacturer’s instructions. Two experiments were carried out. In the first experiment, the mini-pumps were removed on day 17, and the mice were sacrificed for histopathological evaluation on day 20. In the second experiment, the mini-pumps were removed on day 17 and replaced with newly filled devices. These mice were then treated for 17 additional days and analyzed on day 37. The mice were treated for a total of 17/34 days, and the duration of the treatment was 20/37 days in total.

### 4.3. Endpoints

A list of the endpoints compared in this study and whether the analysis was carried out on data already published or on data retrospectively obtained from stored samples are presented in Table 1. For the endpoints already published, the methods are briefly summarized to assure that the determinations were consistent across the study. For the endpoints specifically determined for this study, the methods are discussed in detail.

#### 4.3.1. Blood Determinations

In all the studies, blood was collected from the retro-orbital plexus into ethylenediaminetetraacetic acid (EDTA)-coated microcapillary tubes (20–40 µL/sampling). In the case of Aplidin and SB431542, the HCT, plt, and white blood cell (WBC) counts were determined manually. In the case of AVID200, the P-selectin inhibitor and Ruxolitinib alone and in combination, and Reparixin, Htc, Hemoglobin (Hb), Plts, and white blood cell (WBC) counts were evaluated by the same accredited commercial provider (Plaisant Laboratory, Rome, Italy).

#### 4.3.2. Histological and Immunohistochemical Analyses

The femurs and the spleens were fixed in formaldehyde (10% *v*/*v* with neutral buffer); the femurs were treated for 1 h with bone marrow biopsy decalcifying solution (Osteodec Cat. No. 05-03005Q; Bio-Optica, Milan, Italy) and embedded in paraffin. The sections (3 μm) were stained with either H&E (Harris’s Hematoxylin Cat. No. 01HEMH2500; Eosin Cat. No. 01EOY101000; Histo-Line Laboratories, Pantigliate, MI, Italy), Gomori silver or Reticulin staining, and Mallory trichrome staining (Cat. No. 04-040801, 04-040802, and 04-020802, respectively; Bio-Optica, Milan, Italy). Alternatively, the slides were evaluated by using immunohistochemistry with anti-TGF-β1 (Cat. No. sc-130348; Santa Cruz Biotechnology, Santa Cruz, CA, USA), anti-CXCL1 (Cat. No. ab86436; Abcam, Cambridge, UK), anti-CD34 (Cat. No. MAB7100; AbNova, Taipeh, Taiwan) antibodies. The immunoreactions were detected with avidin–biotin immunoperoxidase (Vectastain Elite ABC Kit; Vector Laboratories, Burlingame, CA, USA) and chromogen 3,3′-diaminobenzidine (0.05% *w*/*v*; Cat. No. ACB999; Histo-Line Laboratories). Slides were counterstained with Harris’s Hematoxylin (Histo-Line Laboratories). Images were acquired by using an optical microscope (Eclipse E600; Nikon, Shinjuku, Japan) equipped with an Imaging Source “33” Series USB 3.0 Camera (Cat. No. DFK 33UX264; Bremen, Germany). The reticulin fibers were quantified on 5 different areas/femur/mouse from at least 4 mice per group by using the ImageJ program (version 1.52t) (National Institutes of Health, Bethesda, MD, USA) as previously described [10]. Bone maturation evaluating Mallory trichrome staining was carried out as previously described [17]. The contents of TGF-β1 and CXCL1 were quantified by using the ImageJ program and expressed as the percentages of positive tissue in an area of 0.144 mm^2^, as previously described [10]. To quantify the total area occupied by bone in the femurs, digital images of H&E-stained sections were acquired by using a commercial whole-slide scanner and were analyzed by using the ImageJ program with the threshold setting as described [10].

#### 4.3.3. Immunofluorescence and Confocal Microscopy

Three-micron-thick bone marrow sections were dewaxed in xylene; the antigens were retrieved by treatment with EDTA buffer (pH 8) for 20’ in a pressure cooker (110–120 °C, high pressure) and were incubated with antibodies against CD42b (Cat. No. ab183345; Abcam, Cambridge, UK) and GATA1 (Cat. No. sc-265; Santa Cruz Biotechnology, Santa Cruz, CA, USA) overnight at 4 °C. The primary antibodies were visualized by using secondary antibody goat anti-rat Alexa Fluor 488 (Cat. No. ab150165; Abcam) or goat anti-rabbit Alexa Fluor 555 (Cat. No. ab150078; Abcam). All the sections were counterstained with DAPI (Cat. No. D9542-5MG; Sigma-Aldrich, Darmstadt, Germany), mounted with Fluor-shield histology mounting medium (Cat. No. F6182-10MG; Sigma-Aldrich), and examined by using a Nikon Eclipse Ni microscope equipped with filters appropriate for the fluorochrome to be analyzed. The images were recorded by using a Nikon DS-Qi1Nc digital camera and NIS 190 Elements software BR 4.20.01. The microvessel density was determined by incubating the bone marrow and spleen sections with anti-CD34 (Cat. No. MAB7100; AbNova, Taiwan), Alexa Fluor 568-conjugated donkey anti-rat (Invitrogen, Carlsbad, CA, USA), and Hoechst 33342 (Thermo Fisher Scientific, Waltham, MA, USA). The images were processed and analyzed by using Zen Blue (3.2) software (Carl Zeiss GmbH, Milano, Italy) and the ImageJ program (version 1.52t) (National Institutes of Health, Bethesda, MD, USA).

### 4.4. Statistical Analyses

The majority of the results regarding the effects of the various treatments on the myelofibrotic phenotype of *Gata1*^low^ mice have already been published [17,21,23,26] and are expressed as Delta (fold change) with respect to the values in the corresponding vehicle group. The data were analyzed and plotted by using GraphPad Prism 8.0.2 software (GraphPad Software, San Diego, CA, USA) and were presented as means (±SD) or as box charts, as more appropriate. All the data had a normal distribution, as assessed by the Shapiro–Wilk test. Values between two groups were compared by using the *t*-test, while those among multiple groups were compared by using the one-way ANOVA multiple comparison test, as specified in the legend of the figures. Differences were considered statistically significant with a *p* < 0.05.

## 5. Conclusions

The comparison of the effects of the treatments carried out so far in the *Gata1*^low^ mouse model of MF demonstrated similarity but also variation in the efficacy of the drugs tested in targeting the various abnormalities of the myelofibrotic phenotype of *Gata1*^low^ mice. Splenomegaly was reduced only by treatments which included Ruxolitinib. The drugs which reduced fibrosis more effectively targeted CXCL1 or TGF-β1. Although the drug which inhibited CXCL1 also decreased TGF-β1 content in bone marrow, the platelet counts were increased only by drugs which inhibited TGF-β directly. For the drugs which have already been tested on patients, there was a good correlation between the effects they exerted on the mouse model and those exerted in vivo. These data suggest that future therapies for MF should consider combining JAK1/2 inhibitors to reduce splenomegaly by using drugs targeting RAC1/2, to reduce malignant HSCs and/or the inflammatory milieu (IL-8 or TGF-β), and to reduce fibrosis and increase platelet counts.

## Figures and Tables

**Figure 1 ijms-25-07703-f001:**
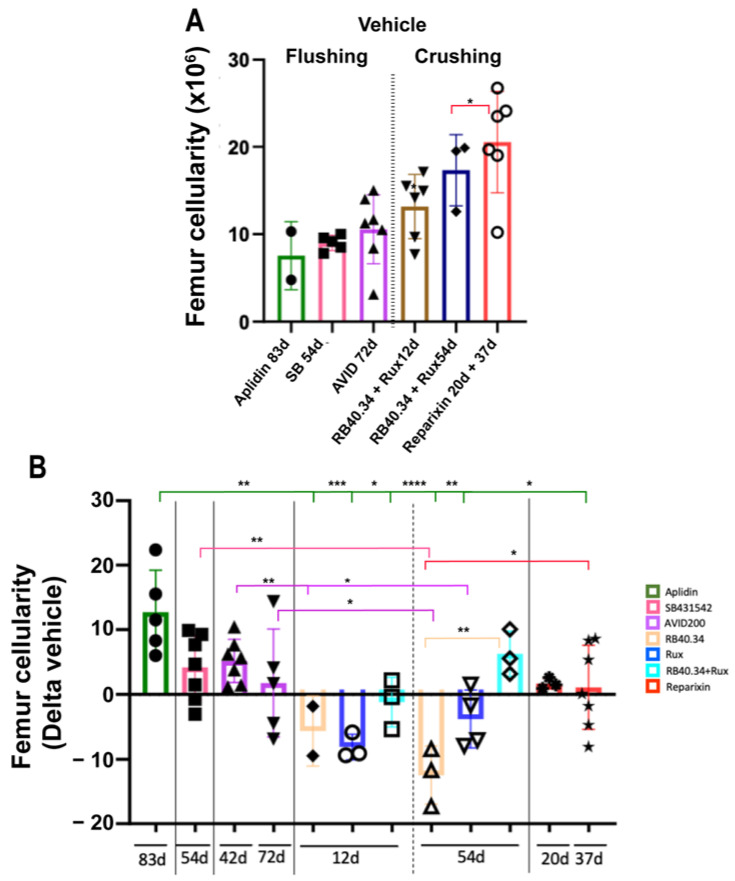
Effects of the various treatments on the total number of cells in the femur. (**A**) Number of cells in the femur in the vehicle groups of all the experiments. Bone marrow cells were recovered either by flushing the femoral cavity or by carefully crushing the whole femur, as indicated. The two sets of values underwent separate statistical analyses. (**B**) Fold changes induced by the various treatments (each color indicates a different treatment) expressed as means (±SD) and as values in individual mice (each mouse corresponds to one symbol). The color code used to indicate the treatments is summarized by the squares on the right and is used consistently in all the figures. Statistical analyses of the fold changes among groups were carried out using One-way ANOVA; significant differences are indicated by asterisks (* = *p* < 0.05, ** = *p* < 0.01, *** = *p* < 0.001, and **** = *p* < 0.0001). The number below the *X*-axis indicates the day of the treatment in which the mice were sacrificed. The original data were published in [17,20,21,23,24,26].

**Figure 2 ijms-25-07703-f002:**
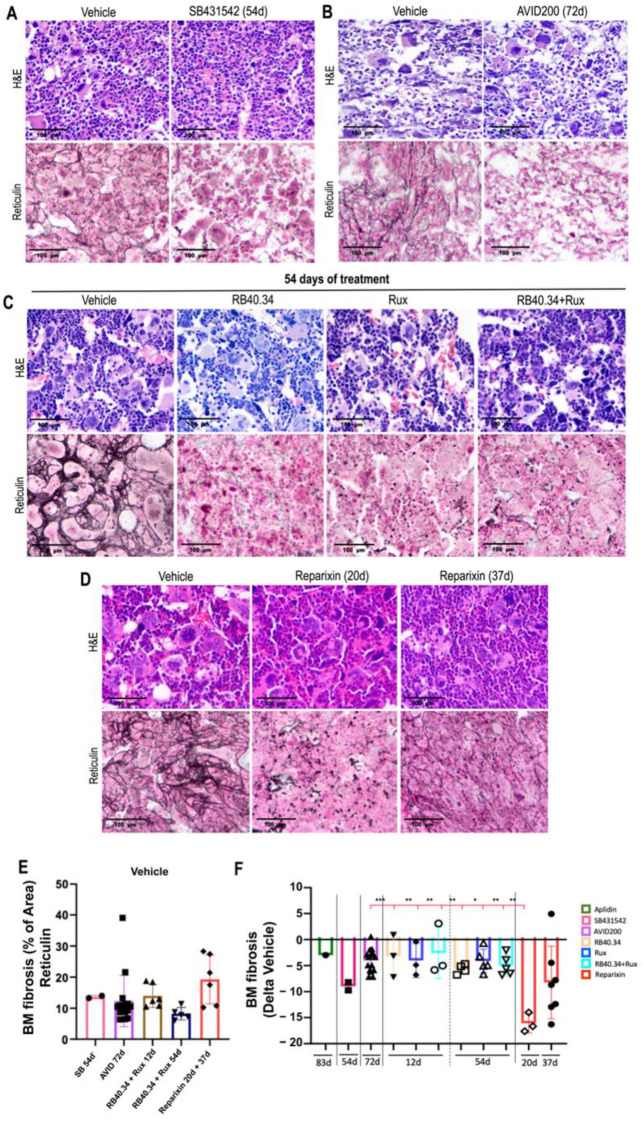
All the treatments effectively reduced fibrosis in the bone marrow of *Gata1*^low^ mice; the drugs which sustained the greatest reductions were Reparixin (20 days) and SB431542 (54 days). (**A**–**D**) Representative images of the H&E and reticulin staining of bone marrow from mice treated with either vehicle or SB431542 (**A**); AVID200 for 42 days (**B**); RB40.34 and Ruxolitinib, alone or in combination, for 54 days (**C**); or Reparixin for 20 and 37 days (**D**) as described. Magnification: 20×. (**E**) Absolute values of bone marrow fibrosis (% of area) of the vehicles in all the experiments and (**F**) changes in bone marrow fibrosis induced by the various treatments are presented as means (±SD) and as values in individual mice (each mouse corresponds to one symbol). The fold changes in the groups were not statistically different when using one-way ANOVA (* = *p* < 0.05, ** = *p* < 0.01, and *** = *p* < 0.001). The original data were published in [17,20,21,23,24,26].

**Figure 3 ijms-25-07703-f003:**
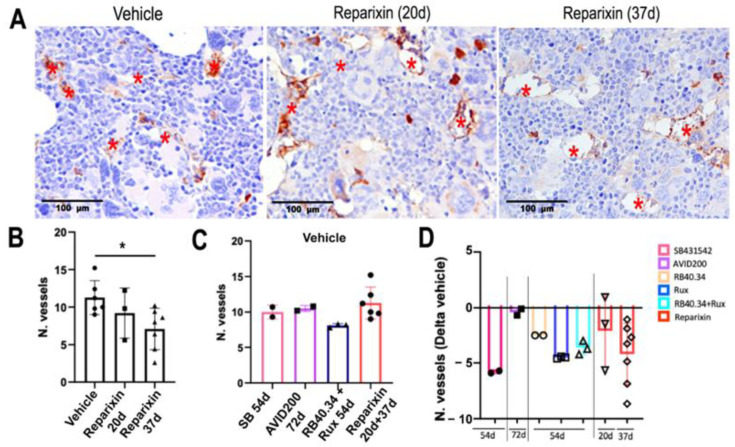
With the exception of AVID200, all the treatments effectively reduced the vessel density of the bone marrow of the *Gata1*^low^ mice. (**A**) Representative immunohistochemical images of the femoral anti-CD34 antibody from the *Gata1*^low^ mice treated with Reparixin. The vessels are indicated by red asterisks. Magnification: 40×. (**B**) Vessel density in the bone marrow of the mice treated with vehicle or Reparixin for 20 and 37 days, as indicated. The number of vessels is the average of those measured in 5 randomly selected photomicrographs per bone marrow section per mouse (area of each photomicrograph = 1.49 mm^2^). The results are presented as means (±SD) and as values in individual mice (each mouse corresponds to one symbol); the statistical analyses were carried out by using one-way ANOVA (* = *p* < 0.05). (**C**) Vessel density in the bone marrow of mice treated with vehicle in all the experiments and (**D**) changes in vessel density induced by the various treatments expressed as Delta of the corresponding vehicle values. Results are presented as means (±SD) and as values in individual mice (each mouse corresponds to one symbol). No statistical differences were found in the Delta induced by the various treatments by one-way ANOVA. The original data were published in [17,21,23,24,26].

**Figure 4 ijms-25-07703-f004:**
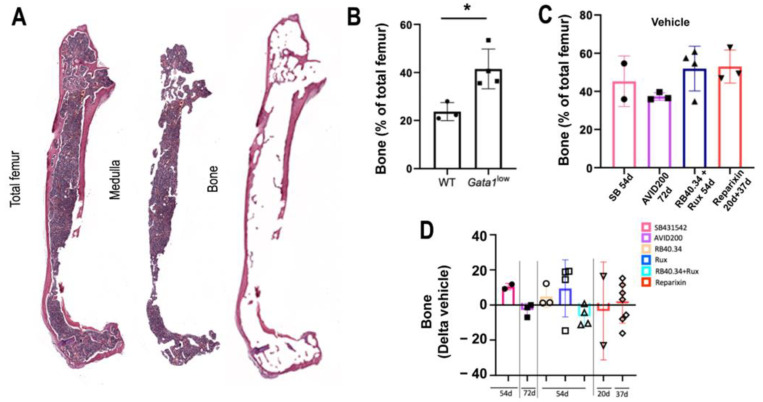
The treatments tested had limited effects on the areas of the femur occupied by bone in *Gata1*^low^ mice. (**A**) Reconstruction of a femur from a representative *Gata1*^low^ mouse treated with vehicle and stained with H&E and depiction of the computer-assisted process used to determine the area of the femur occupied by the medulla and that occupied by bone, respectively. (**B**) Percentages of area with bone tissue in the femur of wild-type and *Gata1*^low^ mice (males, 12 months old). Statistical analyses were carried out by using one-way ANOVA (* = *p* < 0.05). (**C**) Percentages of area with bone tissue in the femur from the vehicle group in all the experiments and (**D**) changes in the area occupied by bone in femurs from mice treated with the various drugs expressed as Delta of the values in the corresponding vehicle group. The results are presented as means (±SD) and as values in individual mice (each mouse corresponds to one symbol). No statistical differences were found among the groups by using one-way ANOVA.

**Figure 5 ijms-25-07703-f005:**
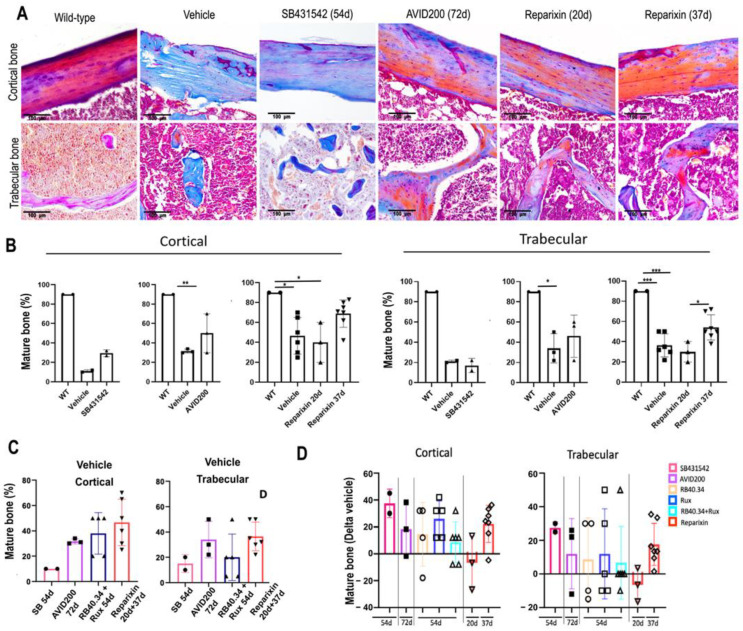
Only AVID200 and Reparixin improved the percentage of mature bone present in the femurs of *Gata1*^low^ mice. (**A**) Representative images of the Mallory trichrome staining of sections of the femurs of wild-type (WT) and *Gata1*^low^ mice treated either with vehicle or with SB431542 (54 days), AVID200 (42 days), and Reparixin (20 and 37 days), as indicated. Magnification: 20×. It should be noted that the effects appear more pronounced in the cortical than in the trabecular areas. (**B**) The quantification of mature bone present in the cortex and trabeculae of the femurs of the WT and *Gata1*^low^ mice treated either with the vehicle or with SB431542 (54 days), AVID200 (42 days), or Reparixin (20 and 37 days). Data are represented as means (±SD) and as values in individual mice (each symbol corresponds to a different mouse). The statistical analysis was carried out by using one-way ANOVA (* = *p* < 0.05, ** = *p* < 0.01, and *** = *p* < 0.001). (**C**) The areas of mature bone present in the cortex and in the trabeculae of the femurs from the vehicle group in all the experiments. (**D**) Fold changes (such as Delta compared with the corresponding vehicle) in areas of mature bone present in the cortex and in the trabeculae of the femurs from *Gata1*^low^ mice treated with the various drugs, as indicated. The results are presented as means (±SD) and as values in individual mice (each mouse corresponds to one symbol). There were no statistically significant differences among the effects exerted by the various treatments by using one-way ANOVA.

**Figure 6 ijms-25-07703-f006:**
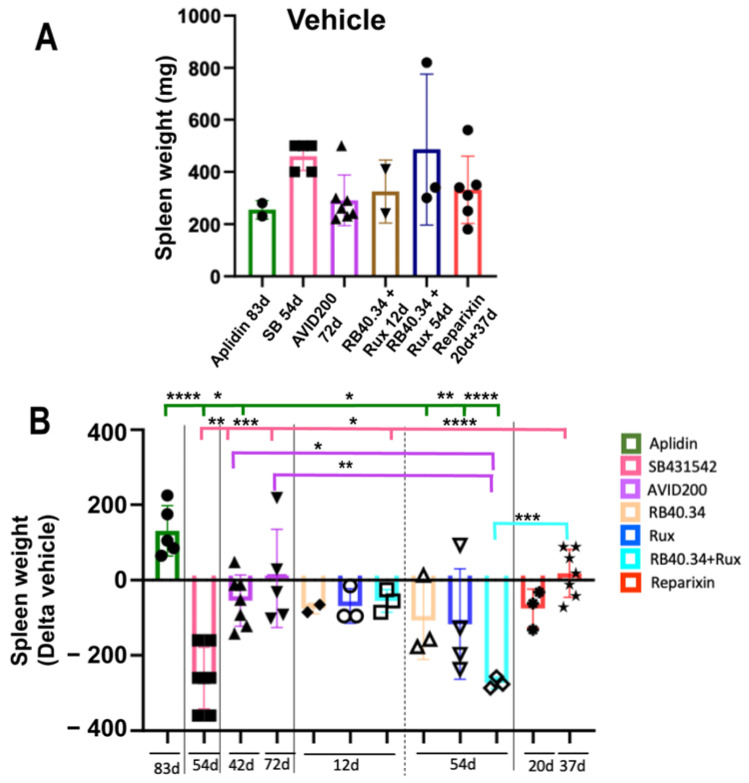
The greatest reductions in spleen weight were obtained by treatment with SB431542 and Ruxolitinib, either alone or in combination with RB40.34. (**A**) Spleen weight in the vehicle of each experiment. (**B**) Changes induced by the various treatments in spleen weight compared with the corresponding vehicle presented as means (±SD) and as values in individual mice (each mouse corresponds to one symbol). Statistical analyses were performed by using one-way ANOVA (* = *p* < 0.05, ** = *p* < 0.01, *** = *p* < 0.001, and **** = *p* < 0.0001). The original data were published in [17,20,21,23,24,26].

**Figure 7 ijms-25-07703-f007:**
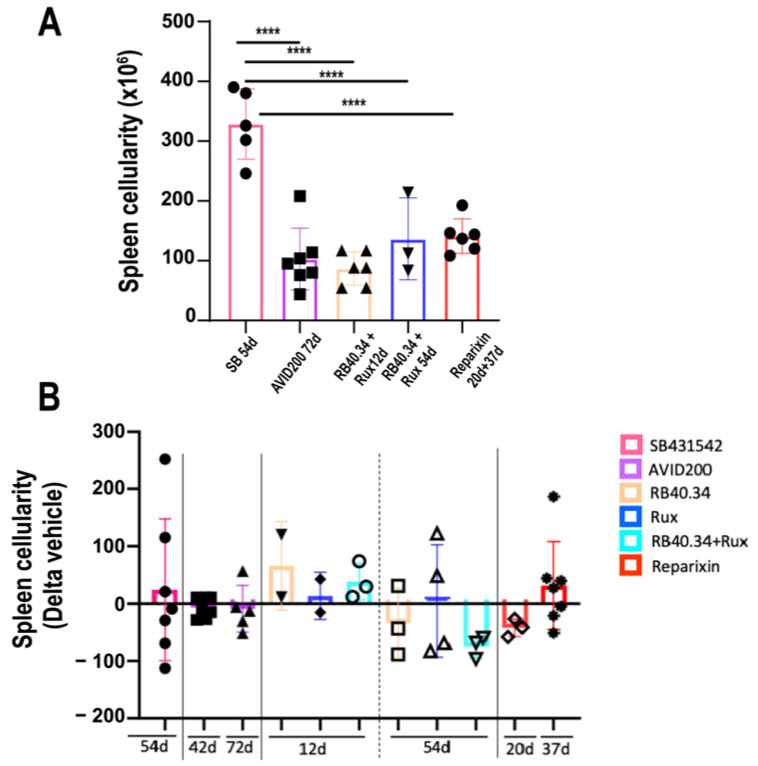
All the treatments tested induced modest changes in the total number of cells present in the spleen. (**A**) Spleen cellularity in the vehicle groups of all the experiments. Only in the case of SB431542 was the cellularity of the spleen from the vehicle group significantly greater than that of the other groups. (**B**) Changes induced by the various treatments in the spleen cellularity compared with the corresponding vehicle. Data are presented as means (±SD) and as values in individual mice (each mouse corresponds to one symbol). The effects induced by the different drugs were not statistically different according to one-way ANOVA (**** = *p* < 0.0001). The original data were published in [17,21,23,24,26].

**Figure 8 ijms-25-07703-f008:**
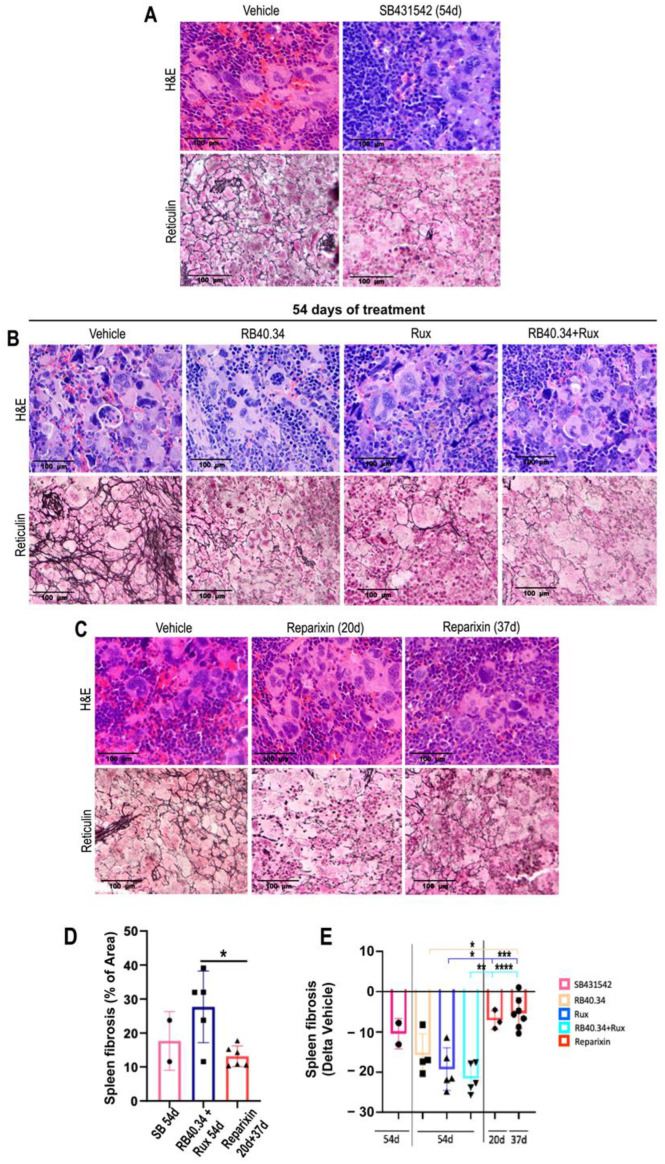
All the treatments reduced fibrosis in the spleen of *Gata1*^low^ mice; the greatest reductions were achieved with Ruxolitinib alone or in combination with RB40.34. (**A**–**C**) Representative unpublished images of the H&E and reticulin staining of spleen sections from mice treated with either the vehicle or SB431542 (54 days) (**A**), RB40.34 or Ruxolitinib alone and in combination (54 days) (**B**), or Reparixin (20 and 37 days) (**C**) as described. Magnification: 40×. (**D**) Level of fibrosis in the spleen in vehicle group of the various experiments and (**E**) fold change in the level of fibrosis in the spleen of the treated group compared with that of the corresponding vehicle group. Data are presented as means (±SD) and as values in individual mice (each mouse corresponds to one symbol). Statistical analysis was carried out by using one-way ANOVA; significant differences are indicated by asterisks (* = *p* < 0.05, ** = *p* < 0.01, *** = *p* < 0.001, and **** = *p* < 0.0001). The original data were published in [17,23,24,26].

**Figure 9 ijms-25-07703-f009:**
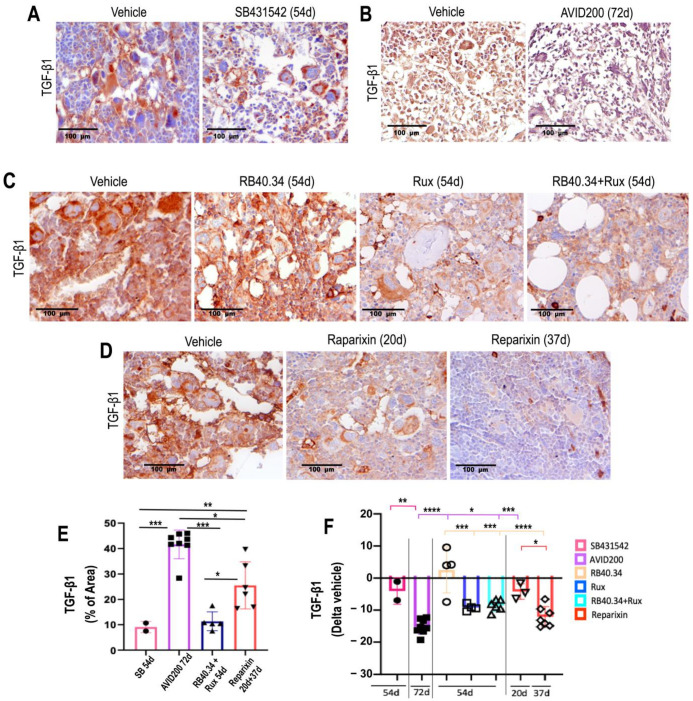
The majority of the treatments tested reduced TGF-β1 content in bone marrow from *Gata1*^low^ mice. (**A**–**D**) Representative images of the immunohistochemistry analyses with the antibody against TGF-β1 of bone marrow sections from mice treated with either the vehicle or SB431542 for 54 days (**A**), AVID200 for 42 days (**B**), RB40.34 or Ruxolitinib alone and in combination for 54 days (**C**), or Reparixin for 20 and 37 days (**D**), as described. Magnification: 40×. (**E**) Levels of TGF-β1 in bone marrow of the vehicle group in each experiment. Data are represented as means (±SD) of the percentage of the area positive for staining and as values in individual mice (each mouse corresponds to one symbol) and (**F**) fold changes in the levels of TGF-β1 with respect to the corresponding vehicle groups (each mouse corresponds to one symbol). The statistical analysis was carried out by using one-way ANOVA; the statistically significant differences are indicated by asterisks (* = *p* < 0.05, ** = *p* < 0.01, *** = *p* < 0.001, and **** = *p* < 0.0001). The original data were published in [17,21,23,24,26].

**Figure 10 ijms-25-07703-f010:**
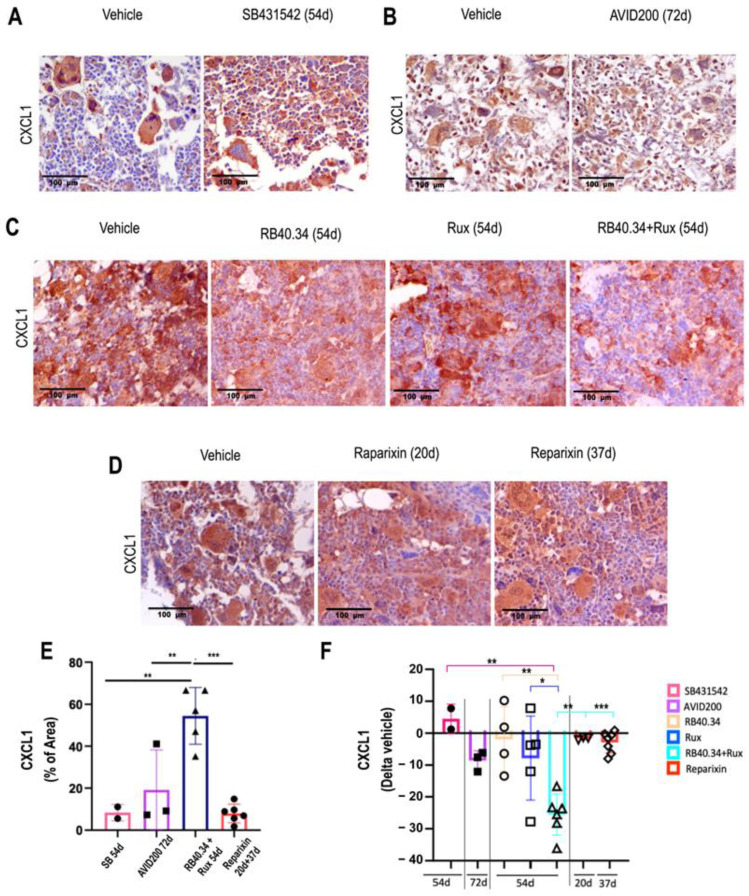
Only RB40.34 in combination with Ruxolitinib reduced CXCL1 content in bone marrow from *Gata1*^low^ mice. (**A**–**E**) Representative images of bone marrow sections from mice treated with either the vehicle or SB431542 for 54 days (**A**), AVID200 for 42 d (**B**), RB40.34 or Ruxolitinib or their combination for 54 days (**C**), or Reparixin for 20 and 37 days (**D**), stained with the antibody against CXCL1, as described. Magnification: 40×. (**E**) Levels of CXCL1 in the bone marrow of the vehicle group in each experiment, represented as means (±SD) of the percentage of the area positive for the staining and as values in individual mice (each mouse corresponds to one symbol) and (**F**) CXCL-1 content in the treated animals presented as fold change with respect to the corresponding vehicle group. The results are presented as means (±SD) and as values in individual mice (each mouse corresponds to one symbol). Statistical analysis among the groups was carried out by using one-way ANOVA; statistically significant differences are indicated by asterisks (* = *p* < 0.05, ** = *p* < 0.01, and *** = *p* < 0.001). Similar data were published in [17,20,21,23,24,26].

**Figure 11 ijms-25-07703-f011:**
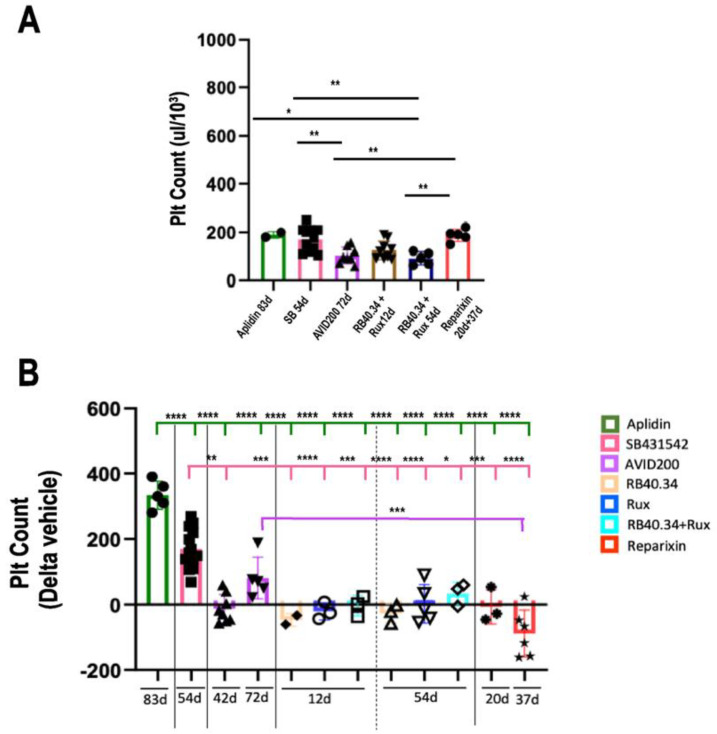
Aplidin and inhibitors of the TGF-β signal (SB431542 and AVID200) increased the platelet counts in *Gata1*^low^ mice. (**A**) Platelet counts in the vehicle in each experiment (each mouse corresponds to one symbol). It should be noted that in spite of being statistically different, the platelet numbers observed in the different vehicles were, for the most part, low. (**B**) The platelet counts were expressed as Delta of the values in the respective vehicle groups and are presented as means (±SD) and as values per individual mouse (each mouse corresponds to one symbol). Statistical analysis of the fold changes among the groups was carried out by using one-way ANOVA; statistically significant differences are indicated by asterisks (* = *p* < 0.05, ** = *p* < 0.01, *** = *p* < 0.001, and **** = *p* < 0.0001). The original data were published in [17,20,21,23,24,26].

**Figure 12 ijms-25-07703-f012:**
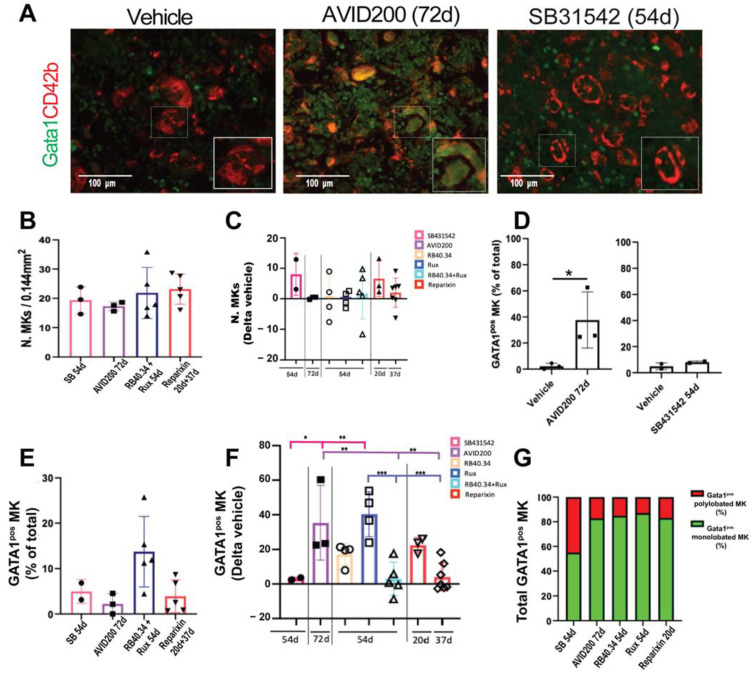
Treatment with the TGF-β inhibitors, Ruxolitinib, or Reparixin increased GATA1 content in the MKs in bone marrow from *Gata1*^low^ mice; however, only in the case of the TGF-β inhibitor SB431542, the increase in the GATA1 content was associated with a mature morphology. (**A**) Representative immunofluorescence analyses with the megakaryocyte marker CD42b and an antibody against GATA1 of bone marrow sections from the femur of a wild-type mouse and from mice treated with either the vehicle or the TGF-β inhibitor AVID200 (42d) or SB431542 (54d), as controls. Magnification: 40× scale bars. (**B**) Frequency of CD42b-positive cells in the vehicle groups from all the experiments. Data are presented as means (±SD) and as values in individual mice and are the average number of CD42-positve cells detected in 5 randomly selected photomicrographs (1.49 mm^2^) per bone marrow section per mouse. No statistically significant difference was detected by using one-way ANOVA. (**C**) Frequency of cD42b-positive cells in bone marrow from mice subjected to the various treatments expressed as Delta values of those in the corresponding vehicle groups. (**D**) Frequency of CD42b-positive cells positive for GATA1 in the vehicles and treated mice in new experiments with AVID200 and SB431542. Statistical analyses were carried out by using one-way ANOVA (* = *p* < 0.05). (**E**) Frequency of CD42b-positive cells positive for GATA1 in the vehicle groups from all the experiments. The results are presented as means (±SD) and as values in individual mice. (**F**) Frequency of CD42b-positive cells positive for GATA1 in mice undergoing the various treatments expressed as Delta values of that of the corresponding vehicle group. The results are presented as means (±SD) and as values in individual mice (each mouse corresponds to one symbol). Statistical analysis of the fold changes across the groups were carried out by using one-way ANOVA; the statistically significant groups are indicated by asterisks (* = *p* < 0.05, ** = *p* < 0.01 and *** = *p* < 0.001). (**G**) Frequency of MKs positive for GATA1 containing monolobated or polylobated nuclei among the different experimental groups. Some of the original data were published in [17,21,23,24,26].

**Figure 13 ijms-25-07703-f013:**
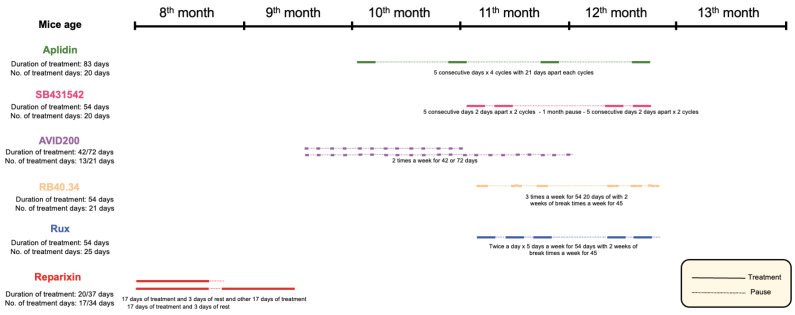
Scheme of all treatments investigated in this study. The timeline of each experiment with the age of the mice at the beginning of the experiment and on the day of their sacrifice for histopathological evaluation is indicated. The total number of days in which each drug was administered and the entire duration of each treatment are also indicated. The continuous lines indicate the days when the drugs were administered, while the dotted lines represent the pause between the sequential cycles of drug administration. The original data were published in [17,20,21,23,24,26].

**Table 1 ijms-25-07703-t001:** Summary of all the endpoints compared in this study and details as to whether the comparison was carried out using data which had been already published or data retrospectively obtained byh using samples stored in the tissue bank. The tissue bank did not contain samples from the Aplidin study.

Parameters	Published	Performed for This Study	Not Available
**Femur**			
**Cellularity (total cell counts of a femur)**	All treatments		
**Fibrosis (reticulin staining)**	Aplidin	Confirmed ex novo for all the other drugs	
**Vessel density (IHC for CD34+ staining)**	SB431542, AVID200, RB40.34, and Rux	Reparixin	Aplidin
**Bone area (computer-assisted evaluation of track images)**		SB431542, AVID200, RB40.34, Rux, and Reparixin	Aplidin
**Bone maturation (Mallory trichrome staining)**		SB431542, AVID200, RB40.34, Rux, and Reparixin	Aplidin
**TGF-β1/CXCL1 content (IHC with specific antibodies)**		SB431542, AVID200, RB40.34, Rux, and Reparixin	Aplidin
**Spleen**			
**Weight**	All treatments		
**Cellularity**	All treatments		Aplidin
**Fibrosis (reticulin staining)**		SB431542, AVID200, RB40.34, Rux, and Reparixin	Aplidin
**Thrombopoiesis**			
**Platelet counts**	All treatments		
**Total MK number in BM (IF with CD42b)**		SB431542, AVID200, RB40.34, Rux, and Reparixin	Aplidin
**GATA1 content in MKs (double IF for GATA1 and CD42b)**		SB431542, AVID200, RB40.34, Rux, and Reparixin	Aplidin

**Table 2 ijms-25-07703-t002:** The summary of the *p*-values between the endpoints observed in the treated mice and those observed in the correspondent vehicle in all the experiments included in this study. The red boxes indicate data already published, the green boxes indicate data from the new dedicated experiments, and the gray boxes indicate that the data are not available. Values not statistically significant are indicated with ns. The original data were published in [17,20,21,23,24,26].

Treatment	Femur Cellularity (*p*-Value vs. Vehicle)	BM Fibrosis (*p*-Value vs. Vehicle)	No. of Vessels (*p*-Value vs. Vehicle)	Bone Area (*p*-Value vs. Vehicle)	Mature Cortical Bone (*p*-Value vs. Vehicle)	Mature Trabecular Bone (*p*-Value vs. Vehicle)	Spleen Weight (*p*-Value vs. Vehicle)	Spleen Cellularity (*p*-Value vs. Vehicle)	Fibrosis Spleen (*p*-Value vs. Vehicle)	TGF-β1 (*p*-Value vs. Vehicle)	CXCL1 (*p*-Value vs. Vehicle)	Platelet Count (*p*-Value vs. Vehicle)	GATA + MKs (*p*-Value vs. Vehicle)
**Aplidin (83d)**	***p* < 0.05**	***p* < 0.05**	***p* < 0.05**	\	\	\	ns	\	\	***p* < 0.05**	\	***p* < 0.05**	***p* < 0.05**
**SB431542 (54d)**	***p* < 0.05**	***p* < 0.05**	***p* < 0.05**	ns	ns	ns	***p* < 0.05**	***p* < 0.05**	***p* < 0.05**	***p* < 0.05**	ns	***p* < 0.0001**	ns
**AVID200 (42d)**	***p* < 0.05**	\	\	\	\	\	ns	ns	\	\	\	***p* < 0.05**	\
**AVID200 (72d)**	***p* < 0.05**	***p* < 0.001**	***p* < 0.001**	ns	ns	ns	ns	ns	ns	***p* < 0.001**	ns	***p* < 0.01**	***p* < 0.05**
**RB40.34 (12d)**	ns	ns	\	\	\	\	ns	ns	\	\	\	ns	\
**RB40.34 (54d)**	ns	***p* < 0.05**	***p* < 0.001**	ns	ns	ns	ns	ns	***p* < 0.05**	ns	ns	ns	ns
**Rux (12d)**	ns	ns	\	\	\	\	ns	ns	\	\	\	ns	\
**Rux (54d)**	ns	ns	***p* < 0.001**	ns	ns	ns	ns	ns	***p* < 0.001**	***p* < 0.05**	ns	ns	***p* < 0.0001**
**RB40.34 + Rux (12d)**	ns	ns	\	\	\	\	ns	ns	\	\	\	ns	\
**RB40.34 + Rux (54d)**	***p* < 0.05**	***p* < 0.05**	***p* < 0.001**	ns	ns	ns	ns	ns	***p* < 0.001**	***p* < 0.05**	***p* < 0.01**	ns	ns
**Reparixin (20d)**	ns	***p* < 0.05**	ns	ns	ns	ns	ns	ns	***p* < 0.05**	***p* < 0.05**	ns	ns	***p* < 0.001**
**Reparixin (37d)**	ns	ns	***p* < 0.05**	ns	ns	ns	ns	ns	ns	***p* < 0.0001**	ns	ns	ns

**Table 3 ijms-25-07703-t003:** Summary of comparative qualitative and quantitative differences in efficacy of treatments investigated.

Endpoints	Most Effective Treatments (in Order of Efficacy)
**Femur**	
**Cellularity**	Aplidin
**Fibrosis**	Reparixin 20 days
**Vessel density**	Reparixin 37 days SB431542 54 days
**Bone area**	RB40.34 + Rux 54 days
**Mature cortical bone**	SB431542 54 days Rux 54 days Reparixin 37 days
**Mature trabecular bone**	SB431542 54 days Reparixin 37 days
**TGF-β1**	AVID200 Reparixin 37 days
**CXCL1**	RB40.34 + Rux 54 days
**Spleen**	
**Weight**	SB431542 54 days RB40.34 + Rux 54 days
**Cellularity**	RB40.34 + Rux 54 days
**Fibrosis**	Rux 54 days RB40.34 + Rux 54 days
**Thrombopoiesis**	
**Platelet counts**	Aplidin SB431542 54 days AVID200
**Total MK number in BM**	None
**GATA1 content in MKs**	AVID200, Rux 54 days, and Reparixin 20 days but only SB431542 in the polylobated cells

## Data Availability

The data for each mouse are available upon request.

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
