# Peer review of "The Variation in the Traits Ameliorated by Inhibitors of JAK1/2, TGF-β, P-Selectin, and CXCR1/CXCR2 in the Gata1low Model Suggests That Myelofibrosis Should Be Treated by These Drugs in Combination"

_ijms, 2024, doi:10.3390/ijms25147703_

Round 1

Reviewer 1 Report

Comments and Suggestions for Authors

Review Migliaccio

In the present work Gobbo and colleagues provide an extensive overview of the effects of different therapeutic strategies on myelofibrosis features developed by Gata1low mice. This interesting and impressive work provide an in vivo rational for the possible use of these drugs for the treatment of MF patients in combination with Ruxolitinib. Nevertheless, I have some concerns that need to be addressed before the work is published.

Major

It is not clear what the authors mean when in the X axis of bar graphs, it is reported 20d for Aplidin, indeed as reported by the authors, mice received drug or vehicle for five consecutive days for four cycles 21 days apart, this means that the treatment lasted for a total of 104 days. What does 20d stands for? SB431542 was injected i.p “for 2 cycles of 5 consecutive days 2 day apart, rested for 1 month, and then treated for 2 additional cycles”, treatment lasted 58d, what does “59d-71d” stands for? How long does AVID200 treatment lasted? How long does the treatment with RB40.34 and/or Ruxolitinib lasted? It is not clear why it is reported a 12 day outcome when in M&M section it is told that RB40.34 was given for 45 days and the i.p.. Can the authors be more accurate in defining the duration of treatments in “4.2 Mice treatments” paragraph?

As regards the different end-points, it is not always clear whether the differences with vehicle for each treatment are statistically significant. In my opinion this point is crucial for the understanding and critical evaluation of the presented results. If authors can not include in the present work the comparison because it has already been published in previous pubblications (e.g reference # 20) they should at least cite this result in Results section with the appropriate reference. This applies to figures 1 (total number of femur cells), 2 (bone marrow fibrosis, as reported in table 1 it was evaluated/confirmed ex-novo for all the drugs apart from Aplidin), figure 4 (bone area), figure 8 (spleen fibrosis), figure 9 (bone marrow TGF-beta content), figure 10 (bone marrow CXCL1 content), figure 12 (total MK numbers and GATA1+ MKs in the bone marrow). When it is possible, authors should include (at least in supplemental materials) a bar plot showing the comparison between vehicle and treated groups with the appropriate statistical analysis, similarly to what they have done for Reparixin treatment in Figure 3B.

In the present study authors decided to compare treatment groups that significantly differ in terms of mice age (e.g. 8 months old mice were treated with Reparixin for 17 or 37 days while 12 months old mice were treated with AVID200 for 15 or 37 days), route of administration, vehicle, duration of treatment and experimental design (number of cycles, number of injections and interruption duration among cycles). Apart from the possible off targets effects that were excluded by the authors due to the absence of significant adverse events, do the authors believe that these additional factors may have influenced their results? Can the authors discuss on this?

According to Figure 3C AVID200 was not able to reduce vessels number compared with vehicle. Please correct line 138 – 139 “All the treatments were equally potent in reducing the vessel density in the bone marrow from Gata1low mice (Figure 3C).” and Figure 3 title. Please correct also Discussion line 456.

It is not clear what the Y axis of Figure 4C represents. Did the authors reported the difference between the percentage of bone area in treated groups and vehicles, or the normalized difference compared with the total bone area in vehicle mice? Indeed, it is shown, for Reparixin 20d a variation that exceeds -20, which is ambiguous considering that the mean image area covered by bone in Gata1low mice do not exceed 8% according to Figure 4B. Please, adjust figure caption in accordance to the answer.

As shown in Figure 5B, no treatment restored bone ossification of cortical and trabecular areas to the level of WT animals, except for a few animals in Reparixin 37d group. I believe that this conclusion (196-198) “The results indicate that all of them improved bone ossification restoring the levels of mature lamellae (stained in red) present both in the cortical and trabecular areas of the Gata1low femurs up to those found in wild-type mice” is a bit overstated. Please tone down this conclusion.

Author says that in Gata1low mice “microenvironmental bioavailability of TGF-β is associated with fibrosis”. Does the author may have the opportunity to test if there is a direct correlation between the reduction in TGF-β BM levels and the reduction of fibrosis in mice treated with the different drugs? It seems that the reduction in TGF-β is more pronounced in AVID200 treated mice compared with SB431542 and in mice treated with Reparixin for 37 days compared with those treated for 20days. At the same time the reduction in BM fibrosis is more pronounced in SB431542 treated animals and mice treated with Reparixin for 20 days, can you discuss this result?

Did the authors check TGF-β microenvironmental bioavailability in the spleen of mice? Does it correlate with the observed fibrosis reduction?

Can the author be more precise in defining Y axis title of panels 9E and 10E. How was the TGF-beta and CXCL1 content measured? Did the authors measured total cytokines content in the bone marrow, or it is reported the MK cytokines content as reported in materials and methods (lines 606 – 608)?

According to Figure 12 SB431542 treated mice display a very low expression level of GATA1, and the frequency of GATA1+ megakaryocytes do not change compared with vehicle treated mice. Nevertheless, it was observed that the frequency of GATA1+ polylobated megakaryocytes is increased compared with other treatments and platelets count is dramatically increased (Figure 11A). These two results are apparently in contrast because, as authors says, “the maturation of the MK, the cells responsible for their (platelets) production, is hampered by the reduced expression of GATA1”. Can the author comment on this? Can SB431542 exert its function independently from GATA1 expression?

Is the difference in distribution of polylobated and monolobated megakaryocytes among treatments statistically significant, can the authors perform statistical analysis on this?

Minor

Can the authors provide a better resolution for images included in Figures 2, Figure 9E, Figure 10E, Figure 11, Figure 12?

Please change the order of tables numeration, in the present version table 2 precedes table 1. In Table 1 please substitute Reparexin with Reparexin.

I suggest an English revision since some typos are present.

Comments on the Quality of English Language

I suggest an English revision since some typos are present.

Author Response

Replay to Reviewer 1

In the present work Gobbo and colleagues provide an extensive overview of the effects of different therapeutic strategies on myelofibrosis features developed by Gata1low mice. This interesting and impressive work provide an in vivo rational for the possible use of these drugs for the treatment of MF patients in combination with Ruxolitinib. Nevertheless, I have some concerns that need to be addressed before the work is published.

Au: We thank this reviewer for his/her kind words of appreciation and for the constructive comments. We found these comments excellent and well suited to increase the clarity of the discussion and the rigor and transparency of the presentation of the data. Individual comments were addressed as indicated below.

Major

  • It is not clear what the authors mean when in the X axis of bar graphs, it is reported 20d for Aplidin, indeed as reported by the authors, mice received drug or vehicle for five consecutive days for four cycles 21 days apart, this means that the treatment lasted for a total of 104 days. What does 20d stands for? SB431542 was injected i.p “for 2 cycles of 5 consecutive days 2 day apart, rested for 1 month, and then treated for 2 additional cycles”, treatment lasted 58d, what does “59d-71d” stands for? How long does AVID200 treatment lasted? How long does the treatment with RB40.34 and/or Ruxolitinib lasted? It is not clear why it is reported a 12 day outcome when in M&M section it is told that RB40.34 was given for 45 days and the i.p.. Can the authors be more accurate in defining the duration of treatments in “4.2 Mice treatments” paragraph?

Au: We thank the reviewer for this comment. The confusion noted by this reviewer arose from an inconsistent use in the X axis of “number of days of treatment” versus “duration of the treatment” among the different experiments. The X axis was revised in order to indicate “duration of the treatment” in all the cases (see the X axes of the new figures). To increase clarity, we have prepared a new figure (Figure 13) which describes the duration of the various treatments, the number of days the mice had actually received the drugs as well as other technical information such as age of the mice at the beging and end of the treatments).

  • As regards the different end-points, it is not always clear whether the differences with vehicle for each treatment are statistically significant. In my opinion this point is crucial for the understanding and critical evaluation of the presented results. If authors can not include in the present work the comparison because it has already been published in previous pubblications (e.g reference # 20) they should at least cite this result in Results section with the appropriate reference. This applies to figures 1 (total number of femur cells), 2 (bone marrow fibrosis, as reported in table 1 it was evaluated/confirmed ex-novo for all the drugs apart from Aplidin), figure 4 (bone area), figure 8 (spleen fibrosis), figure 9 (bone marrow TGF-beta content), figure 10 (bone marrow CXCL1 content), figure 12 (total MK numbers and GATA1+ MKs in the bone marrow). When it is possible, authors should include (at least in supplemental materials) a bar plot showing the comparison between vehicle and treated groups with the appropriate statistical analysis, similarly to what they have done for Reparixin treatment in Figure 3B.

Au: The comparison between the effects of treatment versus vehicle was presented only for those data which have been generated as part of this manuscript. The relevant statistically significant differences between the treatment and vehicle group which had been already published were discussed in the text. However, this reviewer rightly points out that these differences were never systematically presented. In view of this comment, we have added a summary table (new Table 2) with the p-values of the treatments vs vehicles for all the end-points (absolute efficacy) discussed in the manuscript. This Table clarifies which drug was ineffective to start with. The Table is presented just before that of the summary of the relative efficacy among treatments (now Table 3) so that the reader may easily compare absolute vs relative efficacy, as requested.  

  • In the present study authors decided to compare treatment groups that significantly differ in terms of mice age (e.g. 8 months old mice were treated with Reparixin for 17 or 37 days while 12 months old mice were treated with AVID200 for 15 or 37 days), route of administration, vehicle, duration of treatment and experimental design (number of cycles, number of injections and interruption duration among cycles). Apart from the possible off targets effects that were excluded by the authors due to the absence of significant adverse events, do the authors believe that these additional factors may have influenced their results? Can the authors discuss on this?

Au: to clarify possible effects of age, we added a new Figure (Figure 13) which indicates the age of the mice at the beginning and the end of each treatment. In addition, we increased the clarity and the rigor of the comparison by adding to all the figures a comparison of the values observed in the vehicles used to normalize the fold changes in all the experiments. With very few exceptions, the new Figures clarify that the values observed in the vehicle groups are not statistically different across experiments providing support that eventual difference in efficacy is unlikely to be due to differences in the stage of the disease progression of the animals included in the experimental groups, as discussed on page 18, lines 427-430 of the revised manuscript. In addition, we have included a cautionary note that, although we do not believe this to be the case, differences in vehicle (DMSO or saline) and route of administration (i.p., i.v, or gauvage) may have represented confounding factors when comparing data cross experiments (page 18, lines 430-433 of the revised discussion).

  • According to Figure 3C AVID200 was not able to reduce vessels number compared with vehicle. Please correct line 138 – 139 “All the treatments were equally potent in reducing the vessel density in the bone marrow from Gata1lowmice (Figure 3C).” and Figure 3 title. Please correct also Discussion line 456.

Au: We have corrected the capture of Figure 3C, the abstract, the presentation of the data and their discussion to clarify that: “with the exception of AVID200, all the treatments were equally potent in reducing microvessel density”, As suggested.

  • It is not clear what the Y axis of Figure 4C represents. Did the authors reported the difference between the percentage of bone area in treated groups and vehicles, or the normalized difference compared with the total bone area in vehicle mice? Indeed, it is shown, for Reparixin 20d a variation that exceeds -20, which is ambiguous considering that the mean image area covered by bone in Gata1low mice do not exceed 8% according to Figure 4B. Please, adjust figure caption in accordance to the answer.

Au: The caption of the Y axes of Figure 4C and 4D was revised for clarity. The new capture reads: Bone (percent of total femur).

  • As shown in Figure 5B, no treatment restored bone ossification of cortical and trabecular areas to the level of WT animals, except for a few animals in Reparixin 37d group. I believe that this conclusion (196-198) “The results indicate that all of them improved bone ossification restoring the levels of mature lamellae (stained in red) present both in the cortical and trabecular areas of the Gata1lowfemurs up to those found in wild-type mice” is a bit overstated. Please tone down this conclusion.

Au: We tuned down the conclusion by revising this paragraph as follows: The results indicate that AVID200 and Reparixin improved bone ossification and increased somewhat the levels of mature lamellae (stained in red) present both in the cortical and trabecular areas of the Gata1low femurs (Figure 5A,B). However, the lack of significance among groups in terms of Delta values (Figure 5C, D) indicate that the effects of AVID200 and Reparixin on bone maturation were overall modest (see page 8, lines 212-216, and page 22, lines 541-543) .

  • Author says that in Gata1lowmice “microenvironmental bioavailability of TGF-β is associated with fibrosis”. Does the author may have the opportunity to test if there is a direct correlation between the reduction in TGF-β BM levels and the reduction of fibrosis in mice treated with the different drugs? It seems that the reduction in TGF-β is more pronounced in AVID200 treated mice compared with SB431542 and in mice treated with Reparixin for 37 days compared with those treated for 20days. At the same time the reduction in BM fibrosis is more pronounced in SB431542 treated animals and mice treated with Reparixin for 20 days, can you discuss this result?

Au: The hypothesis raised by this reviewer that the bioavailability of TGF-beta and the levels of fibrosis in the bone marrow are correlated is very interesting. Unfortunately, this analysis is not feasible because the two end-points were not tested on consecutive bone marrow sections.

  • Did the authors check TGF-β microenvironmental bioavailability in the spleen of mice? Does it correlate with the observed fibrosis reduction?

Au: We thank the reviewer to raise this important point. The bioavailability of TGF-beta in the bone marrow and spleen from Gata1low mice with respect to control was tested in ref. 17. This paper found that both the bone marrow and spleen from the mutant mice expressed levels of TGF-beta 1.2-fold greater than normal. This paper also characterized the downstream TGF-beta pathways activated in the two organs and how they were affected by treatment with SB431542. The signature of the bone marrow included activation of osteoblast differentiation, apoptosis and G1 arrest and ubiquitin mediated proteolysis. By contrast, the signature of the spleen included only apoptosis and G1 arrest. Treatment with the TGF-beta inhibitor normalized the TGF-beta signature of the bone marrow but did not affect that of the spleen which continued to show activation of apoptosis and G1 arrest and revealed additional abnormalities in the ubiquitin mediated proteolysis pathway. These data enforce our conclusion that differences in cell populations responsible for fibrosis, rather than TGF-beta bioavailability, are responsible for the differences in the efficacy of the various drugs to reduce the fibrosis in the two organs. We have clarified our thoughts on the first paragraph on page 21 of the revised manuscript.

  • Can the author be more precise in defining Y axis title of panels 9E and 10E. How was the TGF-beta and CXCL1 content measured? Did the authors measured total cytokines content in the bone marrow, or it is reported the MK cytokines content as reported in materials and methods (lines 606 – 608)?

Au: We clarified the Y axis of panels 9e and 10e by adding “The content of TGF-β1 and CXCL1 was quantified with ImageJ program and is expressed as percentage of positive tissue in an area of 0.144 mm2, as previously described” (page 25, lines 688-689 of the revised M&M).

  • According to Figure 12 SB431542 treated mice display a very low expression level of GATA1, and the frequency of GATA1+ megakaryocytes do not change compared with vehicle treated mice. Nevertheless, it was observed that the frequency of GATA1+ polylobated megakaryocytes is increased compared with other treatments and platelets count is dramatically increased (Figure 11A). These two results are apparently in contrast because, as authors says, “the maturation of the MK, the cells responsible for their (platelets) production, is hampered by the reduced expression of GATA1”. Can the author comment on this? Can SB431542 exert its function independently from GATA1 expression?

Au: Again, another excellent comment from this reviewer which is supported by data present in the literature. In fact, published data indicate that inhibition of TGF-beta may promote the maturation of a subpopulation of MK (the niche MK) which express high levels of GATA2. In addition, John Crispino published some time ago that high levels of GATA2 may rescue the defective maturation of megakaryocytes with low Gata1 content (Huang et al Mol Cell Biol 2009 Sep;29(18):5168-80.  doi: 10.1128/MCB.00482-09. Epub 2009 Jul 20). We are well aware of the possibility raised by this reviewer, but we have decided not to discuss not too raise the perception of excessive speculation. The experiments to test this hypothesis are under way and will be part of a separate study.

  • Is the difference in distribution of polylobated and monolobated megakaryocytes among treatments statistically significant, can the authors perform statistical analysis on this?

Au: We did not perform the statistical analyses because one of the experimental groups has only two data points. Sorry no enough material in the tissue bank.  

Minor

  • Can the authors provide a better resolution for images included in Figures 2, Figure 9E, Figure 10E, Figure 11, Figure 12?

Au: Our Illustrator files for these Figures are high quality but for some reason when we include the Figures in the word document resolution is lost. High quality figures are available to the Journal because we have independently uploaded them with the submission. IJMS will ensure that they are included in the published version of the manuscript, if accepted. IJMS is aware of this challenge and will be pleased to provide to the reviewer the quality of the Figures is he/she would like to check them. 

  • Please change the order of tables numeration, in the present version table 2 precedes table 1. In Table 1 please substitute Reparexin with Reparexin.

Au: We apologize with the reviewer for the confusion. Table 1 was first mentioned on page 2 and shown later on with the Material and Methods where we felt it properly belongs. To prevent further confusion, Table 1 is now present on page 2.

  • I suggest an English revision since some typos are present.

Au: In view of this comment, the manuscript will be revised by a professional provided for a fee by the editorial office before publication.

Replay to Reviewer 1

In the present work Gobbo and colleagues provide an extensive overview of the effects of different therapeutic strategies on myelofibrosis features developed by Gata1low mice. This interesting and impressive work provide an in vivo rational for the possible use of these drugs for the treatment of MF patients in combination with Ruxolitinib. Nevertheless, I have some concerns that need to be addressed before the work is published.

Au: We thank this reviewer for his/her kind words of appreciation and for the constructive comments. We found these comments excellent and well suited to increase the clarity of the discussion and the rigor and transparency of the presentation of the data. Individual comments were addressed as indicated below.

Major

  • It is not clear what the authors mean when in the X axis of bar graphs, it is reported 20d for Aplidin, indeed as reported by the authors, mice received drug or vehicle for five consecutive days for four cycles 21 days apart, this means that the treatment lasted for a total of 104 days. What does 20d stands for? SB431542 was injected i.p “for 2 cycles of 5 consecutive days 2 day apart, rested for 1 month, and then treated for 2 additional cycles”, treatment lasted 58d, what does “59d-71d” stands for? How long does AVID200 treatment lasted? How long does the treatment with RB40.34 and/or Ruxolitinib lasted? It is not clear why it is reported a 12 day outcome when in M&M section it is told that RB40.34 was given for 45 days and the i.p.. Can the authors be more accurate in defining the duration of treatments in “4.2 Mice treatments” paragraph?

Au: We thank the reviewer for this comment. The confusion noted by this reviewer arose from an inconsistent use in the X axis of “number of days of treatment” versus “duration of the treatment” among the different experiments. The X axis was revised in order to indicate “duration of the treatment” in all the cases (see the X axes of the new figures). To increase clarity, we have prepared a new figure (Figure 13) which describes the duration of the various treatments, the number of days the mice had actually received the drugs as well as other technical information such as age of the mice at the beging and end of the treatments).

  • As regards the different end-points, it is not always clear whether the differences with vehicle for each treatment are statistically significant. In my opinion this point is crucial for the understanding and critical evaluation of the presented results. If authors can not include in the present work the comparison because it has already been published in previous pubblications (e.g reference # 20) they should at least cite this result in Results section with the appropriate reference. This applies to figures 1 (total number of femur cells), 2 (bone marrow fibrosis, as reported in table 1 it was evaluated/confirmed ex-novo for all the drugs apart from Aplidin), figure 4 (bone area), figure 8 (spleen fibrosis), figure 9 (bone marrow TGF-beta content), figure 10 (bone marrow CXCL1 content), figure 12 (total MK numbers and GATA1+ MKs in the bone marrow). When it is possible, authors should include (at least in supplemental materials) a bar plot showing the comparison between vehicle and treated groups with the appropriate statistical analysis, similarly to what they have done for Reparixin treatment in Figure 3B.

Au: The comparison between the effects of treatment versus vehicle was presented only for those data which have been generated as part of this manuscript. The relevant statistically significant differences between the treatment and vehicle group which had been already published were discussed in the text. However, this reviewer rightly points out that these differences were never systematically presented. In view of this comment, we have added a summary table (new Table 2) with the p-values of the treatments vs vehicles for all the end-points (absolute efficacy) discussed in the manuscript. This Table clarifies which drug was ineffective to start with. The Table is presented just before that of the summary of the relative efficacy among treatments (now Table 3) so that the reader may easily compare absolute vs relative efficacy, as requested.  

  • In the present study authors decided to compare treatment groups that significantly differ in terms of mice age (e.g. 8 months old mice were treated with Reparixin for 17 or 37 days while 12 months old mice were treated with AVID200 for 15 or 37 days), route of administration, vehicle, duration of treatment and experimental design (number of cycles, number of injections and interruption duration among cycles). Apart from the possible off targets effects that were excluded by the authors due to the absence of significant adverse events, do the authors believe that these additional factors may have influenced their results? Can the authors discuss on this?

Au: to clarify possible effects of age, we added a new Figure (Figure 13) which indicates the age of the mice at the beginning and the end of each treatment. In addition, we increased the clarity and the rigor of the comparison by adding to all the figures a comparison of the values observed in the vehicles used to normalize the fold changes in all the experiments. With very few exceptions, the new Figures clarify that the values observed in the vehicle groups are not statistically different across experiments providing support that eventual difference in efficacy is unlikely to be due to differences in the stage of the disease progression of the animals included in the experimental groups, as discussed on page 18, lines 427-430 of the revised manuscript. In addition, we have included a cautionary note that, although we do not believe this to be the case, differences in vehicle (DMSO or saline) and route of administration (i.p., i.v, or gauvage) may have represented confounding factors when comparing data cross experiments (page 18, lines 430-433 of the revised discussion).

  • According to Figure 3C AVID200 was not able to reduce vessels number compared with vehicle. Please correct line 138 – 139 “All the treatments were equally potent in reducing the vessel density in the bone marrow from Gata1lowmice (Figure 3C).” and Figure 3 title. Please correct also Discussion line 456.

Au: We have corrected the capture of Figure 3C, the abstract, the presentation of the data and their discussion to clarify that: “with the exception of AVID200, all the treatments were equally potent in reducing microvessel density”, As suggested.

  • It is not clear what the Y axis of Figure 4C represents. Did the authors reported the difference between the percentage of bone area in treated groups and vehicles, or the normalized difference compared with the total bone area in vehicle mice? Indeed, it is shown, for Reparixin 20d a variation that exceeds -20, which is ambiguous considering that the mean image area covered by bone in Gata1low mice do not exceed 8% according to Figure 4B. Please, adjust figure caption in accordance to the answer.

Au: The caption of the Y axes of Figure 4C and 4D was revised for clarity. The new capture reads: Bone (percent of total femur).

  • As shown in Figure 5B, no treatment restored bone ossification of cortical and trabecular areas to the level of WT animals, except for a few animals in Reparixin 37d group. I believe that this conclusion (196-198) “The results indicate that all of them improved bone ossification restoring the levels of mature lamellae (stained in red) present both in the cortical and trabecular areas of the Gata1lowfemurs up to those found in wild-type mice” is a bit overstated. Please tone down this conclusion.

Au: We tuned down the conclusion by revising this paragraph as follows: The results indicate that AVID200 and Reparixin improved bone ossification and increased somewhat the levels of mature lamellae (stained in red) present both in the cortical and trabecular areas of the Gata1low femurs (Figure 5A,B). However, the lack of significance among groups in terms of Delta values (Figure 5C, D) indicate that the effects of AVID200 and Reparixin on bone maturation were overall modest (see page 8, lines 212-216, and page 22, lines 541-543) .

  • Author says that in Gata1lowmice “microenvironmental bioavailability of TGF-β is associated with fibrosis”. Does the author may have the opportunity to test if there is a direct correlation between the reduction in TGF-β BM levels and the reduction of fibrosis in mice treated with the different drugs? It seems that the reduction in TGF-β is more pronounced in AVID200 treated mice compared with SB431542 and in mice treated with Reparixin for 37 days compared with those treated for 20days. At the same time the reduction in BM fibrosis is more pronounced in SB431542 treated animals and mice treated with Reparixin for 20 days, can you discuss this result?

Au: The hypothesis raised by this reviewer that the bioavailability of TGF-beta and the levels of fibrosis in the bone marrow are correlated is very interesting. Unfortunately, this analysis is not feasible because the two end-points were not tested on consecutive bone marrow sections.

  • Did the authors check TGF-β microenvironmental bioavailability in the spleen of mice? Does it correlate with the observed fibrosis reduction?

Au: We thank the reviewer to raise this important point. The bioavailability of TGF-beta in the bone marrow and spleen from Gata1low mice with respect to control was tested in ref. 17. This paper found that both the bone marrow and spleen from the mutant mice expressed levels of TGF-beta 1.2-fold greater than normal. This paper also characterized the downstream TGF-beta pathways activated in the two organs and how they were affected by treatment with SB431542. The signature of the bone marrow included activation of osteoblast differentiation, apoptosis and G1 arrest and ubiquitin mediated proteolysis. By contrast, the signature of the spleen included only apoptosis and G1 arrest. Treatment with the TGF-beta inhibitor normalized the TGF-beta signature of the bone marrow but did not affect that of the spleen which continued to show activation of apoptosis and G1 arrest and revealed additional abnormalities in the ubiquitin mediated proteolysis pathway. These data enforce our conclusion that differences in cell populations responsible for fibrosis, rather than TGF-beta bioavailability, are responsible for the differences in the efficacy of the various drugs to reduce the fibrosis in the two organs. We have clarified our thoughts on the first paragraph on page 21 of the revised manuscript.

  • Can the author be more precise in defining Y axis title of panels 9E and 10E. How was the TGF-beta and CXCL1 content measured? Did the authors measured total cytokines content in the bone marrow, or it is reported the MK cytokines content as reported in materials and methods (lines 606 – 608)?

Au: We clarified the Y axis of panels 9e and 10e by adding “The content of TGF-β1 and CXCL1 was quantified with ImageJ program and is expressed as percentage of positive tissue in an area of 0.144 mm2, as previously described” (page 25, lines 688-689 of the revised M&M).

  • According to Figure 12 SB431542 treated mice display a very low expression level of GATA1, and the frequency of GATA1+ megakaryocytes do not change compared with vehicle treated mice. Nevertheless, it was observed that the frequency of GATA1+ polylobated megakaryocytes is increased compared with other treatments and platelets count is dramatically increased (Figure 11A). These two results are apparently in contrast because, as authors says, “the maturation of the MK, the cells responsible for their (platelets) production, is hampered by the reduced expression of GATA1”. Can the author comment on this? Can SB431542 exert its function independently from GATA1 expression?

Au: Again, another excellent comment from this reviewer which is supported by data present in the literature. In fact, published data indicate that inhibition of TGF-beta may promote the maturation of a subpopulation of MK (the niche MK) which express high levels of GATA2. In addition, John Crispino published some time ago that high levels of GATA2 may rescue the defective maturation of megakaryocytes with low Gata1 content (Huang et al Mol Cell Biol 2009 Sep;29(18):5168-80.  doi: 10.1128/MCB.00482-09. Epub 2009 Jul 20). We are well aware of the possibility raised by this reviewer, but we have decided not to discuss not too raise the perception of excessive speculation. The experiments to test this hypothesis are under way and will be part of a separate study.

  • Is the difference in distribution of polylobated and monolobated megakaryocytes among treatments statistically significant, can the authors perform statistical analysis on this?

Au: We did not perform the statistical analyses because one of the experimental groups has only two data points. Sorry no enough material in the tissue bank.  

Minor

  • Can the authors provide a better resolution for images included in Figures 2, Figure 9E, Figure 10E, Figure 11, Figure 12?

Au: Our Illustrator files for these Figures are high quality but for some reason when we include the Figures in the word document resolution is lost. High quality figures are available to the Journal because we have independently uploaded them with the submission. IJMS will ensure that they are included in the published version of the manuscript, if accepted. IJMS is aware of this challenge and will be pleased to provide to the reviewer the quality of the Figures is he/she would like to check them. 

  • Please change the order of tables numeration, in the present version table 2 precedes table 1. In Table 1 please substitute Reparexin with Reparexin.

Au: We apologize with the reviewer for the confusion. Table 1 was first mentioned on page 2 and shown later on with the Material and Methods where we felt it properly belongs. To prevent further confusion, Table 1 is now present on page 2.

  • I suggest an English revision since some typos are present.

Au: In view of this comment, the manuscript will be revised by a professional provided for a fee by the editorial office before publication.

Reviewer 2 Report

Comments and Suggestions for Authors

Authors organized samples collected from several previous studies,  performed new analyses on bone fibrosis and bone marrow megakaryocytes, and presented data from previous reports as well as from new analyses.  Here are a few specific comments for authors to consider to further improve the manuscript.

Specific comments

1. Authors compounded previously published data with data newly generated from previous-saved samples, resulting in a report of 12 Figures and 2 Tables. This length report gives an impression of data piling lack of focus on data organization/presentation. Authors should make an effort to focus on presenting data from new analyses, probably organizing new data into 7 figures according to the 7 items listed in Table 1 (very good Table!) under the “performed for this study” column. In the view of this reviewer, there is no need to present data already presented in previous reports (such as the platelet data in current figure 1). Instead, authors should discuss the previously presented data in the discussion along with new data.

2. Including a Table as a panel in a figure to document statistical significance levels (as in several Figures) is a bad way for data organization/presentation. Significant statistical difference could be displayed more directly within the dot/bar plots.       

3. Since authors compared samples collected from various previous studies under different experimental conditions (such as age of animals when treatment started, length of treatment, time/age when samples were collected etc), authors should clearly disclose these experimental differences in the results or figure legend at least once so that readers could make their own judgement when referring to data comparing efficacies of different treatments.

4. Manuscript should not focus on the argument that combination therapy should be used for the treatment of PMF. Two obvious reasons based on the view of this reviewer: 1) Current manuscript did not add any stronger argument on top of what had been reported in reference 26 concerning the superiority of combination therapy (Rux + RB40.34) over monotherapy in the pre-clinical Gata1-low mouse model (improved marrow cellularity, reduced fibrosis and enhanced hematopoiesis).    2) Monotherapy was very effective in murine model but much less effective in the clinic, raising the issue of translatability of pre-clinical studies.  Analyzing samples collected from multiple monotherapy studies conducted on the same pre-clinical model, as authors so did in the current study, is helpful but could not resolve the translatability issue concerning therapeutic efficacy.  

Author Response

Replay to Reviewer 2

Authors organized samples collected from several previous studies,  performed new analyses on bone fibrosis and bone marrow megakaryocytes, and presented data from previous reports as well as from new analyses.  Here are a few specific comments for authors to consider to further improve the manuscript.

Au: We thank this reviewer for the precise summary of our data and for the suggestions to improve the clarity of the presentation. His/Her comments were addressed as specified below.

Specific comments

  1. Authors compounded previously published data with data newly generated from previous-saved samples, resulting in a report of 12 Figures and 2 Tables. This length report gives an impression of data piling lack of focus on data organization/presentation. Authors should make an effort to focus on presenting data from new analyses, probably organizing new data into 7 figures according to the 7 items listed in Table 1 (very good Table!) under the “performed for this study” column. In the view of this reviewer, there is no need to present data already presented in previous reports (such as the platelet data in current figure 1). Instead, authors should discuss the previously presented data in the discussion along with new data.

Au: We thank this reviewer for his/her suggestions. However we respectfully point out to his/her attention that the comparative re-analyses of data already published was greatly appreciated and praised by Reviewer 1. In view of reviewer 1’s comment, we decided to maintain all the data presented in the manuscript.

  1. Including a Table as a panel in a figure to document statistical significance levels (as in several Figures) is a bad way for data organization/presentation. Significant statistical difference could be displayed more directly within the dot/bar plots.

Au: Statistical analysis was color coded for clarity and added to the bar plots as lines with asterisks among groups, as suggested.

  1. Since authors compared samples collected from various previous studies under different experimental conditions (such as age of animals when treatment started, length of treatment, time/age when samples were collected etc), authors should clearly disclose these experimental differences in the results or figure legend at least once so that readers could make their own judgement when referring to data comparing efficacies of different treatments.

Au: This point is very well taken and carefully addressed. As suggested by Reviewer 1, all the Figures were revised to include a panel with the absolute values observed in the vehicle group of all the treatments. In addition, a new table provides a summary of the p-values of the absolute effectiveness of the various treatments to allow assessing differences in absolute and relative efficacy. We have also added a new Figure (Figure 13) that summarizes the age of the mice, the number of treatments and their duration for each drug investigated. in addition, as also suggested by reviewer 1, we discuss the possibility that vehicle and way of administration may represent confounding factors when comparing efficacy across experiments with different drugs.

  1. Manuscript should not focus on the argument that combination therapy should be used for the treatment of PMF. Two obvious reasons based on the view of this reviewer: 1) Current manuscript did not add any stronger argument on top of what had been reported in reference 26 concerning the superiority of combination therapy (Rux + RB40.34) over monotherapy in the pre-clinical Gata1-low mouse model (improved marrow cellularity, reduced fibrosis and enhanced hematopoiesis). 2) Monotherapy was very effective in murine model but much less effective in the clinic, raising the issue of translatability of pre-clinical studies.  Analyzing samples collected from multiple monotherapy studies conducted on the same pre-clinical model, as authors so did in the current study, is helpful but could not resolve the translatability issue concerning therapeutic efficacy.  

Au: We agree with this reviewer that ref 26 had already provided support on the superiority to use Ruxolitinib in combination with drugs that reduce inflammation. We have added this consideration on page 21, lines 499-509 of the revised manuscript. We respectfully, however, point to the reviewer attention that the hypothesis to combine JAK2 with drugs targeting the malignant HSC is new. We have revised the abstract, the Discussion (page 23, lines 597-598) and the Conclusion (page 26, lines 731.734) to emphasize this point. We also agree with this reviewer that analyzing samples collected from multiple monotherapy studies conducted on the same pre-clinical model is helpful but does not fully address the translatability issue concerning therapeutic efficacy. We have included this cautionary note on page 21, lines 506-509 of the revised manuscript.

Round 2

Reviewer 1 Report

Comments and Suggestions for Authors

Authors provided answers to all my questions. I do not have any additional comment. I suggest to accept the manuscript.

Comments on the Quality of English Language

I suggest an English revision.